# CLIP-SVD: Efficient and Interpretable Vision–Language Adaptation via Singular Values

**Taha Koleilat**[*]                                                                    *taha.koleilat@mail.concordia.ca*
*Department of Electrical & Computer Engineering, Concordia University, Montreal, Canada*

**Hassan Rivaz**                                                                          *hassan.rivaz@concordia.ca*
*Department of Electrical & Computer Engineering, Concordia University, Montreal, Canada*

**Yiming Xiao**                                                                           *yiming.xiao@concordia.ca*
*Department of Computer Science & Software Engineering, Concordia University, Montreal, Canada*

**Reviewed on OpenReview:** *https://openreview.net/forum?id=XYy8pwqwMR*

## Abstract

Vision-language models (VLMs) like CLIP have shown impressive zero-shot and few-shot learning capabilities across diverse applications. However, adapting these models to new fine-grained domains remains difficult due to reliance on prompt engineering and the high cost of full model fine-tuning. Existing adaptation approaches rely on augmented components, such as prompt tokens and adapter modules, which could limit adaptation quality, destabilize the model, and compromise the rich knowledge learned during pretraining. In this work, we present **CLIP-SVD**, a *multi-modal* and *parameter-efficient* adaptation framework that applies Singular Value Fine-tuning (SVF) to CLIP, leveraging Singular Value Decomposition (SVD) to modify the internal parameter space of CLIP without injecting additional modules. Specifically, we fine-tune only the singular values of the CLIP parameter matrices to rescale the basis vectors for domain adaptation while retaining the pretrained model. This design enables enhanced adaptation performance using only **0.04%** of the model's total parameters and better preservation of its generalization ability. CLIP-SVD achieves state-of-the-art classification results on 11 natural and 10 biomedical datasets, outperforming previous methods in both accuracy and generalization under few-shot settings. Additionally, we leverage a natural language-based approach to analyze the effectiveness and dynamics of the CLIP adaptation to allow interpretability of CLIP-SVD. Overall, this work provides the first extensive empirical evaluation of SVD-based finetuning in the vision-language model setting. The code and biomedical corpus are publicly available at https://github.com/HealthX-Lab/CLIP-SVD.

## 1 Introduction

Vision-language models (VLMs), such as CLIP (Radford et al., 2021), have demonstrated remarkable versatility and generalization by aligning images and text through large-scale contrastive pretraining. These models enable powerful zero-shot and few-shot capabilities for various applications, but adapting them effectively to downstream tasks remains non-trivial. Full model fine-tuning is often computationally infeasible, while prompt learning strategies (e.g., CoOp (Zhou et al., 2022b) and CoCoOp (Zhou et al., 2022a)) may be limited by heavy dependency on handcrafted or learned text prompts. Adapter-based methods, such as CLIP-Adapter (Gao et al., 2024) and MaPLe (Khattak et al., 2023a), infuse additional modules to improve adaptation quality, but this increases model complexity, lowers inference efficiency, and sometimes degrades zero-shot performance by destabilizing pretrained representations (Khattak et al., 2023b). Recent efforts

---

[*]Corresponding Author

in parameter-efficient fine-tuning (PEFT) aim to overcome these limitations. Among them, Singular Value Fine-Tuning (SVF) (Sun et al., 2022) has emerged as a compelling strategy by modifying only the singular values of model weight matrices without changing the original model. While Sun et al. (2022) and Meng et al. (2024) showed that SVF has promise in CNNs and large language models (LLMs), its application to Transformer-based, multi-modal vision-language models remains underexplored. Furthermore, despite the popularity of CLIP adaptation methods, very few have attempted to interpret the dynamics and effectiveness of model adaptation. Lastly, most CLIP adaptation techniques focus on natural domains alone, with few specialized in biomedical applications (Bie et al., 2024; Koleilat et al., 2025b) due to distinctive visual features and complex clinical descriptions. This creates a gap for a universal strategy that can generalize effectively across natural and biomedical domains without high computational complexity or tailored adjustments (e.g., specialized prompt engineering (Bie et al., 2024; Koleilat et al., 2025b)).

To address the aforementioned challenges, we present **CLIP-SVD**, a parameter-efficient framework that adapts the existing SVF technique to CLIP for unified few-shot learning across natural and biomedical domains. While many relevant methods (Zhou et al., 2022b) primarily focus on the text branch, recent ones (Khattak et al., 2023a) have shown the benefit of adapting image and text branches jointly, at the cost of heavy "add-on" modules. For example, the popular MaPLe requires additional trainable parameters for 2.85% of the CLIP model. In contrast, our approach leverages Singular Value Decomposition (SVD) to decompose the projection weights in CLIP's attention and feedforward layers into corresponding singular values and singular vectors, with only the first fine-tuned for both image and text encoders. We hypothesize that this allows the model to rescale the basis vectors for each downstream task with superior adaptation quality and generalizability. Furthermore, our combination of CLIP's multi-head attention and SVD-based weight adaptation invites an opportunity for a natural language-based paradigm to localize, rank, and semantically "describe" the most significant dynamic shifts in the adapted model. Here, we probe the best text basis to map the semantic meaning of the attention heads (Gandelsman et al., 2023) with the most significant updates through CLIP-SVD, as shown in Table 1 for 16-shot CLIP adaptation on distinct tasks. Compared with previous efforts (Sun et al., 2022) that rely on visual interpretation and/or model weight statistics, this approach offers more intuitive and granular insights into CLIP adaptation. Yet, a related text corpus for analyzing biomedical data is still unavailable.

**Our study has four major contributions**: **First**, we present the first extensive empirical evaluation of Singular Value Fine-Tuning (Sun et al., 2022) in Transformer-based vision-language models (e.g., CLIP and BiomedCLIP), requiring just **0.04%** of the model's total parameters, significantly lower than other multi-modal methods. **Second**, we performed comprehensive validation with 11 natural and 10 biomedical domain datasets, demonstrating CLIP-SVD's superior performance against the state-of-the-art (SOTA) methods in both accuracy and generalization. **Third**, with ranked weight changes associated with our method, we adopted a natural language-facilitated approach to intuitively interpret the effectiveness and dynamics of task-specific CLIP adaptation. **Lastly**, to meet an urgent need for semantic interpretation of attention heads in CLIP for biomedical applications (e.g., analysis of CLIP-SVD), we built the first corpus of biomedical image descriptions.

## 2 Related Works

### 2.1 Parameter-efficient Fine-tuning

Fine-tuning large VLMs is often computationally prohibitive for domain-specific adaptation. Parameter-efficient fine-tuning addresses this by updating only a small subset of parameters while keeping the backbone frozen (Lialin et al., 2023). Selective tuning methods like BitFit (Zaken et al., 2022) adjust only bias terms, while pruning and sparsity techniques can further reduce trainable parameters (Guo et al., 2021; Holmes et al., 2021), though they often compromise robustness in zero-shot settings. Adapter-based tuning offers a more robust alternative by inserting lightweight modules (Rebuffi et al., 2017; Houlsby et al., 2019; Karimi Mahabadi et al., 2021; Chen et al., 2022b; Lian et al., 2022), but can introduce inference latency (Pfeiffer et al., 2021). Popular prompt tuning (Lester et al., 2021; Li & Liang, 2021; Jia et al., 2022) and Low-Rank Adaptation (LoRA) (Hu et al., 2021) provide other PEFT strategies, with LoRA inserting low-rank matrices to minimize overfitting (Li et al., 2018; Aghajanyan et al., 2021). However, these methods

Table 1: Natural language-based interpretations for the top 3 Attention Heads associated with the highest normalized changes (sorted in descending order) in the Output-Value circuit after CLIP adaptation for different datasets. Here, "**L**" denotes layer while "**H**" denotes attention head.

| EuroSAT (Satellite Images) | DTD (Texture Images) |
|---|---|
| (**L10.H0**): Aerial Landscapes & Environments
(**L10.H10**): Mood, Atmosphere & Highlights
(**L11.H0**): Semantic Layout & Spatial Context | (**L8.H6**) Refined Textural Details of Everyday Objects
(**L10.H2**) Cultural & Textural Scenes
(**L9.H4**) Natural Landscapes & Textures |
| SUN397 (Scene Understanding) | UCF101 (Action Recognition) |
| (**L11.H0**): Semantic Layout & Spatial Context
(**L11.H2**): Numbers, Symbols & Temporal Cues
(**L11.H3**): Lifestyle & Tranquil Activities | (**L10.H5**) Action, Emotion & Faces
(**L10.H6**) Organic Flow & Movement
(**L10.H1**) Human Experiences & Objects in Action |
| BUSI (Breast Ultrasound) | BTMRI (Brain MRI) |
| (**L11.H8**): Converging Edges & Cluster Markers
(**L8.H6**): Contour Irregularity & Internal Spread
(**L8.H3**): Radiologic Artifacts & Diffuse Shapes | (**L8.H9**) Scattered Highlights & Artifactual Spots
(**L8.H0**) Focal Markers & Shape Cues
(**L9.H5**) Streaks, Texture, & Soft Borders |
| COVID-QU-Ex (Chest X-ray) | CTKIDNEY (Kidney CT) |
| (**L11.H0**): Signal Voids & Shifts
(**L9.H1**): Ring-Like Structures & Localized Spread
(**L11.H3**): Cross-Lobe Flow & Density Buildup | (**L8.H6**): Contour Irregularity & Internal Spread
(**L8.H3**): Radiologic Artifacts & Diffuse Shapes
(**L8.H10**): Diffuse Zones & Overlapping Shapes |

typically rely on external, randomly initialized modules, risking destabilization and forgetting of original CLIP knowledge (Zhang et al., 2023b; Zhu et al., 2024b; Shuttleworth et al., 2025). Designing PEFT methods that enhance adaptation without compromising pre-trained strengths remains a major goal. Recently, Singular Value Fine-Tuning (SVF) (Sun et al., 2022) has emerged as a promising alternative. Initially applied to CNNs for segmentation with strong results, SVF modifies only the singular values of weight matrices while preserving their directions without introducing new modules. Later, SAM-PARSER (Peng et al., 2024) extended SVF to large vision Transformer models, but limited its application to the vision encoder without fine-tuning query, key, and value (Q, K, V) matrices crucial for cross-modal tasks. Additionally, SVD-based methods have shown effectiveness in LLMs (Meng et al., 2024), but their potential for multi-modal VLM adaptation remains largely unexplored, offering an exciting direction for future research.

## 2.2 Adapting Vision-Language Models

Vision-language models, such as CLIP (Radford et al., 2021) and ALIGN (Jia et al., 2021), have significantly advanced multi-modal learning by aligning image and text embeddings in a shared space using self-supervised contrastive training. These models perform well on general-domain tasks like zero-shot classification and cross-modal retrieval, but their reliance on broad, non-specialized datasets limits their effectiveness in expert domains, such as healthcare, where nuanced visual cues and domain-specific semantics are critical. To address this, recent research has explored adapting VLMs to specialized settings using techniques like prompt learning, which offers a lightweight alternative to full fine-tuning. Methods such as CoOp (Zhou et al., 2022b) and CoCoOp (Zhou et al., 2022a) learn optimized prompts while keeping the VLM backbone frozen, with extensions like MaPLe (Khattak et al., 2023a) and PromptSRC (Khattak et al., 2023b) improving robustness through encoder tuning and self-regularization. Adapter-based strategies like CLIP-Adapter (Gao et al., 2024) and Tip-Adapter (Zhang et al., 2021) modify the visual branch or incorporate support-set features to boost few-shot performance, although they may face optimization hurdles. Enhanced probing methods such as LP++ (Huang et al., 2024) further refine adaptation by balancing modality-specific features with adaptive learning dynamics. In the biomedical domain, adaptations of CLIP like BioViL (Boecking et al., 2022), PubMedCLIP (Eslami et al., 2021), and BiomedCLIP (Zhang et al., 2024) leverage domain-specific corpora to improve relevance, with specific methods bridging general-purpose biomedical VLMs and specialized clinical tasks (Koleilat et al., 2024; 2025a; Spiegler et al., 2025; Rasaee et al., 2025). Yet, these models still struggle with fine-grained clinical understanding (Xu et al., 2024; Zhao et al., 2023). Prompt-learning methods, including XCoOp (Bie et al., 2024) and DCPL (Cao et al., 2024), extend CoOp-style tuning to

medical applications, but often demand relatively large training sets. In comparison, BiomedCoOp (Koleilat et al., 2025b) demonstrates that prompt tuning can preserve generalization across diverse medical tasks even in low-resource conditions. Despite this broad range of techniques, no method has yet achieved robust performance across both natural and biomedical domains.

## 3 Method

### 3.1 CLIP Preliminaries

CLIP consists of a vision encoder $\boldsymbol{E_v}$ and a text encoder $\boldsymbol{E_t}$ that project images and text into a shared embedding space. Given a batch of $B$ images and $C$ distinct classes, image inputs $\boldsymbol{X_v} \in \mathbb{R}^{B \times 3 \times H \times W}$ are RGB images of height $H$ and width $W$, and text inputs $\boldsymbol{X_t} \in \mathbb{R}^{C \times L}$ are tokenized sequences of length $L$, where each sequence serves as a text prompt representing a single class.

The encoders generate modality-specific features:

$$\mathbf{V} = \boldsymbol{E_v}(\boldsymbol{X_v}) \in \mathbb{R}^{B \times D}, \quad \mathbf{T} = \boldsymbol{E_t}(\boldsymbol{X_t}) \in \mathbb{R}^{C \times D} \tag{1}$$

where $D$ is the embedding dimension. Both $\mathbf{V}$ and $\mathbf{T}$ are L2-normalized onto the unit hypersphere.

In zero-shot classification, CLIP matches an image to $C$ class descriptions (e.g., `"a photo of a [CLASS]"`). The probability of assigning image embedding $\hat{\mathbf{V}}$ to class $k$ is:

$$p(Y = k | \hat{\mathbf{V}}, \hat{\mathbf{T}}) = \frac{\exp(\hat{\mathbf{V}}^\top \hat{\mathbf{T}}^{(k)} / \tau)}{\sum_{j=1}^{C} \exp(\hat{\mathbf{V}}^\top \hat{\mathbf{T}}^{(j)} / \tau)} \tag{2}$$

where $\tau$ is a learnable temperature parameter. The predicted class $\hat{k}$ is:

$$\hat{k} = \arg\max_k p(Y = k | \hat{\mathbf{V}}, \hat{\mathbf{T}}) \tag{3}$$

This formulation enables CLIP to generalize to unseen categories by leveraging the alignment between images and natural language descriptions.

### 3.2 Singular Value Decomposition of Weight Matrices

**SVD-Based Decomposition of Pre-trained Weights:** The overall framework of CLIP-SVD is shown in Fig. 1. For our proposed CLIP-SVD technique, we decompose the weight matrices in the **Multi-Head Self-Attention (MHSA)** and **Multi-Layer Perceptron (MLP)** blocks of each Transformer layer in CLIP's text and image encoders with SVD. Specifically, each weight matrix $W$ in the MHSA and MLP blocks can be factorized using SVD as follows:

$$W = USR^\top \tag{4}$$

where $U \in \mathbb{R}^{d \times r}$ is the left singular vector matrix, $S = diag(\lambda_1, \lambda_2, \lambda_3, ..., \lambda_r) \in \mathbb{R}^{r \times r}$ is a diagonal matrix containing the singular values ($\lambda_1 \geq \lambda_2 \geq \cdots \geq \lambda_r \geq 0$) arranged in a descending order, $R \in \mathbb{R}^{m \times r}$ is the right singular vector matrix, and $r = \min(d, m)$ is the rank of $W$. Instead of fully modifying $W$ directly, we **freeze the singular vectors** $U$ **and** $R$ **and fine-tune only the singular values** $\lambda_i$. Note that we adapt the vector of full-rank singular values for our application. We further analyze the effect of different rank configurations in Appendix E.

**Multi-Head Self-Attention Computation:** Each Transformer layer in CLIP-type models applies MHSA using the Query (Q), Key (K), Value (V), and Output (O) projection matrices:

$$W_Q, W_K, W_V \in \mathbb{R}^{D \times d}, \quad W_O \in \mathbb{R}^{d \times D}. \tag{5}$$

Given an input $X \in \mathbb{R}^{B \times L \times D}$, self-attention for the $h^{th}$ head is computed as:

$$Q_h = XW_{Q_h} = X(U_{Q_h}S_{Q_h}R_{Q_h}^\top), K_h = XW_{K_h} = X(U_{K_h}S_{K_h}R_{K_h}^\top) \tag{6}$$

$$Z_h = \text{softmax}\left(\frac{Q_h K_h^\top}{\sqrt{d}}\right)(XW_{V_h}) = \text{softmax}\left(\frac{Q_h K_h^\top}{\sqrt{d}}\right)(XU_{V_h}S_{V_h}R_{V_h}^\top) \tag{7}$$

where $B$ is the batch size, $L$ is the sequence length, $D$ is the embedding dimension, and $d$ is the dimension of each attention head.

For $G$ attention heads of a Transformer layer:

$$Z_{\text{MHSA}} = \text{Concat}(Z_1, \ldots, Z_G)(W_O) = \text{Concat}(Z_1, \ldots, Z_G)(U_O S_O R_O^\top). \tag{8}$$

The output of the multi-head self-attention is then combined with the input via a residual connection:

$$X' = X + Z_{\text{MHSA}}. \tag{9}$$

**Feed-forward Network:** Following the self-attention block, the updated representation $X'$ is passed through a feedforward MLP block ($\{W_{in}, W_{out}\}$), which is also decomposed using SVD. The MLP applies two linear transformations with an activation function in between, and finally, a residual connection is applied:

$$H = \text{ReLU}(X'W_{in}) = \text{ReLU}(X'U_{in}S_{in}R_{in}^\top) \tag{10}$$

$$X'' = H(W_{out}) = H(U_{out}S_{out}R_{out}^\top) \tag{11}$$

$$X_{\text{out}} = X' + X''. \tag{12}$$

With CLIP-SVD, each Transformer layer maintains its original representational capacity by rescaling the singular vectors, thus allowing robust adaptation and retention of pretrained knowledge. At initialization, each pretrained weight matrix undergoes a one-time SVD decomposition, after which the resulting weights are stored and reused across different adaptation settings and tasks (the associated computational cost is detailed in Appendix J). During training and inference, each linear layer operates entirely in its SVD-based form. After decomposing a pretrained weight matrix, the singular vectors are frozen, and only the singular values are updated. The model never reuses the original full weights; instead, every forward pass reconstructs the effective weight from the fixed vectors and the trainable singular values. We do not enforce non-negativity on these values, in line with prior works (Sun et al., 2022); allowing them to change signs is harmless since any sign flip can be absorbed into the corresponding singular vectors without affecting the resulting transformation. If a standard dense weight is ever needed, such as when exporting or merging the adaptation back into the backbone, it can be reconstructed from the stored SVD components without altering any learned parameters. In this way, the factorized representation itself serves as the complete, updated form of each adapted layer.

### 3.3 Semantic interpretation of CLIP-SVD with TextSpan

To understand how our adaptation reshapes CLIP's internal representations, we utilize `TextSpan` (Gandelsman et al., 2023), which aligns text descriptions to each attention head $h$ of layer $l$ in the ViT image encoder to reveal their semantic roles. This is achieved by decomposing the MHSA outputs $x^{l,h}$ into a summation of contributions from image tokens through SVD:

$$x^{l,h} = \sum_{i=0}^{N} x_i^{l,h}, \quad x_i^{l,h} = \alpha_i^{l,h} W_O^{l,h} W_V^{l,h} z_i^{l-1} = \alpha_i^{l,h}(U^{l,h}S^{l,h}V^{l,h\top})z_i^{l-1}, \tag{13}$$

where $\alpha_i^{l,h}$ denotes the attention weight from the class token to token $i$, and $z_i^{l-1}$ is the input token representation. `TextSpan` interprets the semantic roles of heads by projecting candidate text features $\mathbf{T}_{\text{corpus}}$ into the span of $x^{l,h}$:

$$\tilde{\mathbf{T}}^{l,h} = x^{l,h}(x^{l,h^\top}x^{l,h})^{-1}x^{l,h^\top}\mathbf{T}_{\text{corpus}}, \text{with} \quad P^{l,h} = x^{l,h} \cdot \tilde{\mathbf{T}}^{l,h\top}, \tag{14}$$

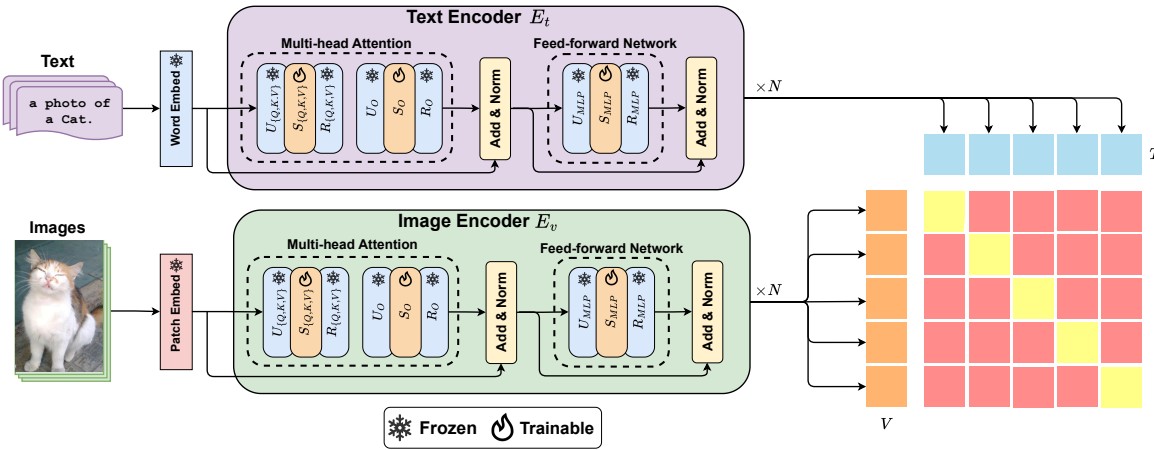

Figure 1: The overall framework of CLIP-SVD. We decompose the Query, Key, Value, and Output projection weights ($W_Q$, $W_K$, $W_V$ and $W_O$) of the MHSA blocks in both vision and text encoders $\boldsymbol{E_v}$ and $\boldsymbol{E_t}$, as well as the linear weights of the Feed-forward Networks ($W_{MLP}$) in all layers. We finetune only the singular values $S$ of the SVD decomposed weights.

and identifies directions that maximize explained variance in $P^{l,h}$. Importantly, this procedure depends only on the span of the singular vectors $U^{l,h}$, not their magnitudes. In contrast, CLIP-SVD finetunes only the singular values $S^{l,h}$:

$$\tilde{x}_i^{l,h} = \alpha_i^{l,h} U^{l,h} \tilde{S}^{l,h} V^{l,h\top} z_i^{l-1} = \alpha_i^{l,h} \sum_j \tilde{s}_j^{l,h} \langle v_j^{l,h}, z_i^{l-1} \rangle u_j^{l,h}, \qquad (15)$$

which preserves the Output-Value (OV) subspace $\text{span}(U^{l,h})$ while reweighting its basis directions through $\tilde{s}_j^{l,h}$. Thus, `TextSpan` and CLIP-SVD are complementary: the former reveals *which* semantic directions are encoded in $\text{span}(U^{l,h})$, while the latter modulates *how much* each direction contributes after adaptation. To quantify these effects, we rank heads by the normalized change in their singular values $S_O^{l,h}$ and $S_V^{l,h}$, summing absolute changes across both matrices. This provides a direct measure of the functional shifts in heads that `TextSpan` already grounds in interpretable text semantics.

## 4 Experiments and Results

### 4.1 Benchmark evaluation settings

**Few-Shot Learning:** To assess the model's performance under limited supervision, we conduct few-shot image classification experiments with varying numbers of labeled examples per class ($K = 1, 2, 4, 8,$ and $16$ shots) to assess the robustness of our method across both natural and biomedical domains. This is critical for evaluating the method's ability to learn effectively from sparse data by obtaining task-specific knowledge while retaining general domain comprehension.

**Base-to-Novel Generalization:** We evaluate the generalizability of CLIP-SVD in natural and biomedical domains, and follow a zero-shot setting, where the datasets are split into base and novel classes for classification tasks. Here, the model is trained only on the base classes in a few-shot setting and evaluated on both base and novel categories. Additionally, we compute the harmonic mean (HM) of both base and novel class prediction accuracies.

**Cross-dataset Evaluation:** To validate the performance of our approach in cross-dataset transfer, we evaluate our ImageNet-trained model directly on other datasets in the natural domain. Consistent with previous methods, our model is trained on all 1000 ImageNet classes in a few-shot manner. Due to the lack of a similar ImageNet-like dataset and large domain shifts across datasets, we didn't perform cross-dataset evaluation for the biomedical domain.

**Datasets:** For the natural domain, we follow Zhou et al. (2022b;a) and evaluate the performance of our method on 11 image classification datasets that cover a wide range of recognition tasks. This includes two generic-objects datasets, ImageNet (Deng et al., 2009) and Caltech101 (Fei-Fei et al., 2004); five fine-grained class-specific datasets, OxfordPets (Parkhi et al., 2012), StanfordCars (Krause et al., 2013), Flowers102 (Nilsback & Zisserman, 2008), Food101 (Bossard et al., 2014), and FGVCAircraft (Maji et al., 2013); a scene recognition dataset SUN397 (Xiao et al., 2010); an action recognition dataset UCF101 (Soomro et al., 2012); a texture dataset DTD (Cimpoi et al., 2014); and a satellite-image dataset EuroSAT (Helber et al., 2019). For the biomedical domain, we follow Koleilat et al. (2025b) and evaluate the performance of our method on 10 diverse medical imaging datasets covering 9 different organs and 8 imaging modalities: Computerized Tomography (CTKidney (Islam et al., 2022)), Endoscopy (Kvasir (Pogorelov et al., 2017)), Fundus Photography (RETINA (Porwal et al., 2018; Köhler et al., 2013)), Histopathology (LC25000 (Borkowski et al., 2019), CHMNIST (Kather et al., 2016)), brain tumor Magnetic Resonance Imaging (BTMRI (Nickparvar, 2021)), Optical Coherence Tomography (OCTMNIST (Kermany et al., 2018)), breast ultrasound (BUSI (Al-Dhabyani et al., 2020)), and chest and knee X-Ray (COVID-QU-Ex (Tahir et al., 2021), KneeXray (Chen, 2018)).

**Implementation Details** We adopt a few-shot training strategy across all experiments, using random sampling for each class. For natural-image tasks, we employ the standard ViT-B/16 CLIP (Radford et al., 2021) model pretrained on 400M image–text pairs from the WebImageText (WIT) corpus, where the vision and text encoders are jointly optimized to align images with natural-language descriptions. For biomedical-domain experiments, we use BiomedCLIP (Zhang et al., 2024), a domain-specialized VLM pretrained on 15M biomedical image–caption pairs from PubMed Central (PMC), combining a ViT-B/16 image encoder with a PubMedBERT text encoder. All competing adaptation baselines, CoOp (Zhou et al., 2022b), CoCoOp (Zhou et al., 2022a), MaPLe (Khattak et al., 2023a), CLIP-LoRA (Zanella & Ben Ayed, 2024), and BiomedCoOp (Koleilat et al., 2025b), are initialized from these same pretrained encoders to ensure a fair comparison. For methods that use prompt tuning, we initialized the prompts with the embedding of "`a photo of a`", while other adaptation techniques were randomly initialized. Each method differs only in its adaptation strategy, whereas CLIP-SVD fine-tunes exclusively the singular values of the weight matrices, preserving the original pretrained geometry of both CLIP and BiomedCLIP. All models are trained with a batch size of 32 using the AdamW optimizer (Loshchilov & Hutter, 2017) with a weight decay of 0.01 on a single NVIDIA A100 GPU (40GB RAM). For natural-domain datasets, the learning rate is set to $5 \times 10^{-4}$ for few-shot classification, $6 \times 10^{-4}$ for base-to-novel evaluations, and $5 \times 10^{-4}$ for cross-dataset transfer. In the biomedical domain, learning rates are tuned per dataset based on validation performance, accounting for task complexity and imaging modality. We report all classification accuracies averaged over three independent runs. Prompt templates and complete hyperparameter configurations are provided in Appendix A and Appendix C, respectively.

## 4.2 Few-shot Evaluation

Our method demonstrates superior performance in few-shot learning across both natural and biomedical domains, as shown in Tables 2 and 3. In the natural domain, CLIP-SVD achieves a +1.00% improvement over the second-best method (CLIP-LoRA) in the 1-shot setting (73.20% vs. 72.20%). Notably, when CLIP-LoRA is constrained to a comparable trainable-parameter budget by reducing its rank from 2 to 1, its few-shot performance slightly degrades as can be seen in Table 2, whereas CLIP-SVD continues to consistently outperform CLIP-LoRA under identical parameter constraints, highlighting superior per-parameter efficiency and the benefits of preserving the pretrained spectral subspace. In the biomedical domain, it surpasses the second-best approach (BiomedCoOp) by +4.28% in the 8-shot setting (73.24% vs. 68.96%). These consistent gains highlight the robustness and effectiveness of our SVD-based tuning strategy across diverse domains.

## 4.3 Base-to-Novel Generalization

Our method demonstrates strong base-to-novel generalization across both natural and biomedical domains, as shown in Tables 4 and 5. In the natural domain, CLIP-SVD improves over MaPLe by +1.67% on base accuracy, +1.06% on novel accuracy, and +1.58% on the harmonic mean, despite MaPLe being approximately 38× more computationally expensive. In the biomedical domain, CLIP-SVD achieves substantial gains over

Table 2: **Evaluation against state-of-the-art techniques for natural domain:** The average classification accuracy (%) obtained from 11 benchmarks derived from 3 sampled support sets for each dataset. The top-performing results are in bold, and the second-best are underlined.

| Method | $K = 1$ | $K = 2$ | $K = 4$ | $K = 8$ | $K = 16$ |
|---|---|---|---|---|---|
| Zero-shot CLIP (Radford et al., 2021) | | | 65.36 | | |
| CoOp (Zhou et al., 2022b) | 68.09 | 70.13 | 73.59 | 76.45 | 79.01 |
| CoCoOp (Zhou et al., 2022a) | 66.95 | 67.63 | 71.98 | 72.92 | 75.02 |
| ProGrad (Zhu et al., 2024a) | 68.20 | 71.78 | 74.21 | 77.93 | 79.20 |
| KgCoOp (Yao et al., 2023a) | 69.51 | 71.57 | 74.48 | 75.82 | 77.26 |
| MaPLe (Khattak et al., 2023a) | 69.27 | 72.58 | 75.37 | 78.89 | 81.79 |
| Linear Probing (Radford et al., 2021) | 45.77 | 56.92 | 66.79 | 73.43 | 78.39 |
| LP++ (Huang et al., 2024) | 70.35 | 72.93 | 75.77 | 77.94 | 80.32 |
| CLIP-Adapter (Gao et al., 2024) | 67.87 | 70.20 | 72.65 | 76.92 | 79.86 |
| Tip-Adapter (Zhang et al., 2021) | 68.89 | 70.42 | 72.69 | 74.41 | 76.44 |
| Tip-Adapter-F (Zhang et al., 2021) | 70.62 | 73.08 | 75.75 | 78.51 | 81.15 |
| GDA (Wang et al., 2024) | 69.39 | 73.09 | 76.24 | 79.71 | 81.70 |
| ProKeR (Bendou et al., 2025) | 71.32 | 73.74 | 76.23 | 79.84 | 82.01 |
| AdaLoRA (Zhang et al., 2023a) | 69.04 | 72.21 | 75.50 | 78.13 | 80.95 |
| TCP (Yao et al., 2024) | 70.63 | 73.59 | 76.07 | 78.39 | 80.98 |
| CLIP-LoRA (rank = 1) (Zanella & Ben Ayed, 2024) | 72.16 | 75.29 | 77.28 | 80.02 | 82.83 |
| CLIP-LoRA (rank = 2) (Zanella & Ben Ayed, 2024) | 72.20 | 75.41 | 77.32 | 80.10 | 82.89 |
| **CLIP-SVD (Ours)** | **73.20** | **76.06** | **78.18** | **80.55** | **82.97** |

Table 3: **Evaluation against state-of-the-art techniques for biomedical domain:** The average classification accuracy (%) obtained from 10 benchmarks derived from 3 sampled support sets for each dataset. The top-performing results are in bold, and the second-best are underlined.

| Method | $K = 1$ | $K = 2$ | $K = 4$ | $K = 8$ | $K = 16$ |
|---|---|---|---|---|---|
| Zero-shot BiomedCLIP (Zhang et al., 2024) | | | 42.38 | | |
| CoOp (Zhou et al., 2022b) | 52.59 | 55.71 | 61.35 | 67.74 | 71.48 |
| CoCoOp (Zhou et al., 2022a) | 50.88 | 53.91 | 57.63 | 63.15 | 67.51 |
| ProGrad (Zhu et al., 2023) | 53.67 | 56.42 | 62.10 | 67.06 | 69.21 |
| KgCoOp (Yao et al., 2023b) | 54.31 | 55.79 | 60.92 | 66.00 | 67.71 |
| Linear Probing (Radford et al., 2021) | 48.91 | 55.82 | 62.12 | 67.33 | 70.81 |
| LP++ (Huang et al., 2024) | 49.27 | 55.88 | 61.30 | 65.48 | 70.09 |
| CLIP-Adapter (Gao et al., 2024) | 45.53 | 44.70 | 45.30 | 46.54 | 48.46 |
| Tip-Adapter (Zhang et al., 2021) | 50.35 | 53.50 | 58.33 | 62.01 | 67.60 |
| Tip-Adapter-F (Zhang et al., 2021) | 52.55 | 54.17 | 62.30 | 68.12 | 68.12 |
| GDA (Wang et al., 2024) | 49.56 | 58.39 | 63.41 | 70.60 | 72.86 |
| ProKeR (Bendou et al., 2025) | 49.40 | 58.84 | 63.72 | 70.98 | 71.86 |
| XCoOp (Bie et al., 2024) | 52.50 | 55.39 | 60.87 | 66.37 | 71.04 |
| DCPL (Cao et al., 2024) | 49.65 | 58.65 | 62.62 | 68.65 | 70.79 |
| CLIP-LoRA (Zanella & Ben Ayed, 2024) | 48.31 | 57.63 | 62.31 | 68.16 | 70.31 |
| MaPLe (Khattak et al., 2023a) | 37.99 | 40.89 | 44.09 | 47.37 | 52.93 |
| BiomedCoOp (Koleilat et al., 2025b) | **56.87** | 59.32 | 64.34 | 68.96 | 73.41 |
| **CLIP-SVD (Ours)** | 56.35 | **62.63** | **68.02** | **73.26** | **76.46** |

BiomedCoOp, improving base accuracy by +4.04%, novel accuracy by +0.41%, and the harmonic mean by +4.21%. These results highlight the robustness and scalability of our SVD-based tuning approach across domains. In addition, they also demonstrate the benefit of multi-modal tuning. The proposed CLIP-SVD enhances generalization without compromising the powerful representations learned during CLIP's pretraining.

Table 4: **Base-to-novel generalization** comparison measured by classification accuracy (%) between CLIP-SVD and SOTA methods on 11 natural domain datasets.

| Acc. | CLIP | CoOp | CoCoOp | KgCoOp | ProGrad | MaPLe | IVLP | GDA | TCP | CLIP-LoRA | CLIP-SVD |
|------|-------|-------|--------|--------|---------|-------|-------|-------|-------|-----------|----------|
| Base | 69.34 | 82.69 | 80.47 | 80.73 | 82.48 | 82.28 | 84.21 | 83.96 | 84.13 | 84.10 | **84.38** |
| Novel | 74.22 | 63.22 | 71.69 | 73.60 | 70.75 | 75.14 | 71.79 | 74.53 | 75.36 | 74.80 | **76.29** |
| HM | 71.70 | 71.66 | 75.83 | 77.00 | 76.16 | 78.55 | 77.51 | 78.72 | 79.51 | 79.18 | **80.13** |

Table 5: **Base-to-novel generalization** comparison measured by classification accuracy (%) between CLIP-SVD and SOTA methods on 10 biomedical domain datasets.

| Acc. | BiomedCLIP | CoOp | CoCoOp | KgCoOp | ProGrad | MaPLe | XCoOp | BiomedCoOp | GDA | DCPL | CLIP-LoRA | CLIP-SVD |
|------|------------|-------|--------|--------|---------|-------|-------|------------|-------|-------|-----------|----------|
| Base | 49.27 | 76.71 | 75.52 | 71.90 | 75.69 | 65.40 | 74.62 | 78.60 | 57.70 | 73.70 | 70.56 | **82.64** |
| Novel | 67.17 | 65.34 | 67.74 | 65.94 | 67.33 | 49.51 | 63.19 | 73.90 | 64.66 | 69.35 | 59.84 | **74.31** |
| HM | 55.23 | 68.80 | 69.11 | 67.22 | 69.86 | 53.10 | 68.43 | 74.04 | 60.98 | 71.46 | 64.76 | **78.25** |

Table 6: **Cross-dataset natural image benchmark** with classification accuracy (%)

| | Source | Target | | | | | | | | | | |
|---|--------|--------|--------|-------------|------------|---------|----------|--------|-------|---------|--------|---------|
| | ImageNet | Caltech101 | OxfordPet | StanfordCars | Flowers102 | Food101 | Aircraft | SUN397 | DTD | EuroSAT | UCF101 | Average |
| CLIP | 66.72 | 92.98 | 89.13 | 65.29 | 71.30 | 86.11 | 24.90 | 62.59 | 44.56 | 47.84 | 66.83 | 65.15 |
| CoOp | 71.51 | 93.70 | 89.14 | 64.51 | 68.71 | 85.30 | 18.47 | 64.15 | 41.92 | 46.39 | 66.55 | 63.88 |
| Co-CoOp | 71.02 | **94.43** | 90.14 | 65.32 | 71.88 | 86.06 | 22.94 | 67.36 | 45.73 | 45.37 | 68.21 | 65.74 |
| KgCoOp | 70.66 | 93.92 | 89.83 | 65.41 | 70.01 | **86.36** | 22.51 | 66.16 | 46.35 | 46.04 | 68.50 | 65.51 |
| ProGrad | **72.24** | 91.52 | 89.64 | 62.39 | 67.87 | 85.40 | 20.61 | 62.47 | 39.42 | 43.46 | 64.29 | 62.71 |
| MaPLe | 70.72 | 93.53 | 90.49 | **65.57** | 72.23 | 86.20 | 24.74 | 67.01 | **46.49** | **48.06** | 68.69 | 66.30 |
| CLIP-SVD | 72.15 | 93.68 | **91.06** | 65.00 | **72.45** | 86.21 | **26.03** | **67.74** | 45.15 | 47.51 | **69.91** | **66.99** |

## 4.4 Cross-dataset Transfer

Table 6 shows that CLIP-SVD achieves the highest average accuracy of 66.99%, slightly outperforming MaPLe's 66.30%. It obtains the best performance on several target datasets, including Aircraft (+1.29%), SUN397 (+0.73%), and UCF101 (+1.22%). These results suggest that CLIP-SVD offers strong cross-dataset transfer capabilities, confirming its potential for effective generalization.

## 4.5 Ablation Experiments: Selective Model Component Fine-tuning

**Effect of Tuning Different Weights:** We ablated CLIP-SVD's components ($W_Q$, $W_K$, $W_V$, $W_O$, and $W_{MLP}$) under a 4-shot setting in natural and biomedical domains (see Table 7). Without adaptation, the accuracy was 65.36% (natural) and 42.38% (biomedical). Adding $W_O$ alone substantially improved performance (75.45% and 62.27%), highlighting its importance. Including $W_{MLP}$ further boosted accuracy (77.78% and 66.40%), showing its role in fine-grained transformation. Adding $W_Q$, $W_K$, or $W_V$ individually on top of $W_O$ and $W_{MLP}$ led to near-identical results in the natural domain, but slight gains in biomedical, up to 67.40% with $W_V$. Using all components yielded the best performance (natural: 78.18% and biomedical: 68.02%), confirming the benefit of full adaptation. Previously, Biderman et al. (2024) observed similar trends when finetuning LLMs in math and coding tasks with LoRA, where a more prominent impact is also seen with adapting MLP than the attention layers.

**Effect of Tuning Image and Text Encoders:** The *right panel* of Fig. 2 illustrates how different input modality tuning setups affect classification accuracy. Multi-modal tuning (text+image) consistently outperforms unimodal cases in both domains. In the natural domain, text-only and image-only achieve 75.78% and 74.31% accuracy, while their combination reaches 78.18%. In the biomedical domain, multi-modal tuning

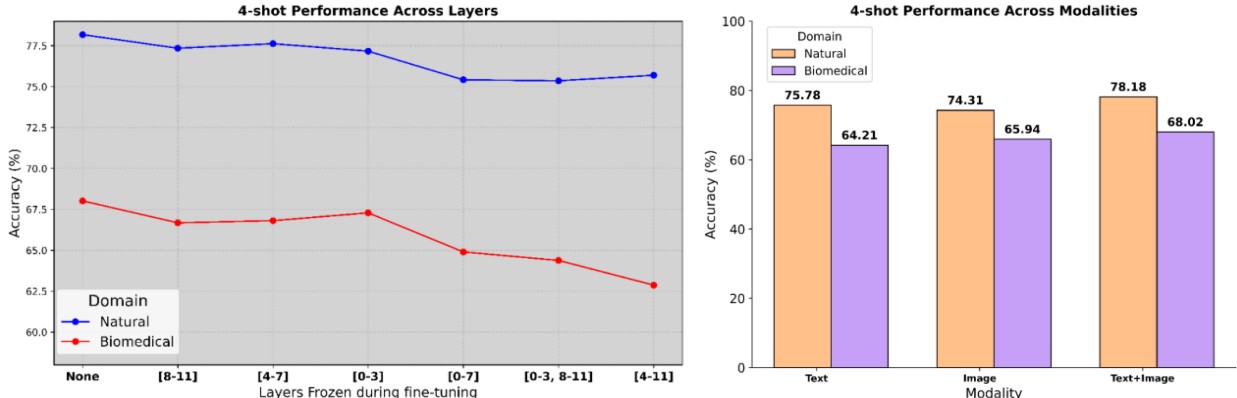

Figure 2: 4-shot performance by freezing certain layers during finetuning (*left*) and by adapting text encoder and/or image encoder (*right*) for natural and biomedical domains.

yields 68.02%, compared to 64.21% (text) and 65.94% (image), suggesting the complementary nature of visual and textual cues, especially valuable in biomedical settings with limited data and higher complexity.

**Effect of Tuning Different Layers:** The *left panel* of Fig. 2 shows how freezing different sets of Transformer layers that are matched in both text and image encoders during SVD-based adaptation affects few-shot accuracy in natural and biomedical domains. In the natural domain, accuracy stays relatively stable, dropping only slightly from 78.18% (all layers adapted) to 77.35% (first four layers frozen) and 75.36% (first and last four frozen), suggesting robust, distributed representations. In contrast, the biomedical domain is more sensitive: accuracy drops from 68.02% (all layers adapted) to 66.68% (last four frozen), 64.38% (first and last four), and 62.87% (top eight frozen), indicating that deeper layers are more critical for capturing domain-specific complexity.

Table 7: Impact of each component of the proposed CLIP-SVD on the 4-shot accuracy (%) of natural and biomedical domain benchmarks.

| $W_Q$ | $W_K$ | $W_V$ | $W_O$ | $W_{MLP}$ | Natural | Biomedical |
|---|---|---|---|---|---|---|
| ✗ | ✗ | ✗ | ✗ | ✗ | 65.36 | 42.38 |
| ✗ | ✗ | ✗ | ✗ | ✓ | 76.69 | 65.27 |
| ✗ | ✗ | ✗ | ✓ | ✗ | 75.45 | 62.27 |
| ✗ | ✗ | ✗ | ✓ | ✓ | 77.78 | 66.40 |
| ✓ | ✓ | ✓ | ✗ | ✗ | 77.15 | 65.35 |
| ✓ | ✓ | ✓ | ✗ | ✓ | 77.83 | 67.50 |
| ✓ | ✓ | ✓ | ✓ | ✗ | 77.88 | 66.74 |
| ✓ | ✗ | ✗ | ✓ | ✓ | 78.12 | 66.99 |
| ✗ | ✓ | ✗ | ✓ | ✓ | 78.12 | 67.09 |
| ✗ | ✗ | ✓ | ✓ | ✓ | 78.11 | 67.40 |
| ✓ | ✓ | ✓ | ✓ | ✓ | **78.18** | **68.02** |

## 4.6 Natural Language-based Interpretation of CLIP-SVD

Natural language offers a powerful but under-explored lens into the conceptual space of VLMs. To gain insights into the impact of CLIP-SVD, we investigate how textual descriptions align with CLIP's internal representations. Besides the natural domain, we present the first **systematic, text-based analysis** of a biomedical VLM at the **attention head level**, focusing on how fine-tuning, such as in BiomedCLIP, reshapes semantic VLM's representations. To support this, similar to Gandelsman et al. (2023), we constructed a new *biomedical caption corpus* of 300 clinically relevant text elements using GPT-4 (Achiam et al., 2023), describing features like contrast, shape, and texture. The correctness and clinical relevance of the generated captions were validated by a human rater, with detailed corpus statistics provided in Appendix B. This enables interpretable alignment between vision and language representations and domain-targeted probing of attention heads. Our framework combines **CLIP-SVD** with `TextSpan` (Gandelsman et al., 2023) to quantify semantic shifts in Output-Value circuits of ViT backbones, focusing on the last four layers (Gandelsman et al., 2023). By ranking the attention heads via singular value shift magnitude from CLIP-SVD and extracting aligned text spans, we gain an intuitive interpretation for the importance of different attention heads during model adaptation and their roles in the finetuned tasks (see Tables 1 and 8). Additional TextSpan alignment score analyses and statistical validation details using LLMs are reported in Appendix G. These analyses highlight how finetuning steers VLMs toward task-relevant, domain-specific understanding. Additionally,

Table 8: Top 3 descriptions returned by TextSpan (Gandelsman et al., 2023) applied to the attention head with the greatest adaptation-related change for different datasets.

| EuroSAT (**L10.H0**) | BUSI (**L11.H8**) | DTD (**L8.H6**) |
| --- | --- | --- |
| Aerial view of an agricultural field | A low-contrast region in a clustered pattern | Collage of textures |
| Image taken in the Namibian desert | A double-density sign suggesting benignity | Close-up of a textured bark |
| Picture taken in the Brazilian rainforest | A solid-cystic component suggesting malignancy | Mesmerizing kinetic sculpture |
| BTMRI (**L8.H9**) | SUN397 (**L11.H0**) | COVID-QU-Ex (**L11.H0**) |
| An area with decreased perfusion in the left hemisphere | Mysterious day scene | A collapsed lung lobe |
| A bright spot artifact in a clustered pattern | Urban rooftop panorama | A low signal-to-noise ratio in the upper lobe |
| A contrast-enhanced region on axial view | A zoomed out photo | A lesion crossing compartments |
| UCF101 (**L10.H5**) | CTKIDNEY (**L8.H6**) | RETINA (**L8.H4**) |
| Dynamic action | An anatomical displacement | An area with decreased perfusion |
| Energetic children | A spiculated margin | A vascular displacement |
| Playful winking facial expression | A zone of tissue infiltration | A vascular structure with sharp borders |

the insights could allow debugging and further refinement (e.g., with prompt engineering) of task/domain-specific CLIP models.

## 5  Discussion

Compared with SVF (Sun et al., 2022) that targeted CNNs and single-modal learning, our CLIP-SVD introduces the first application of SVD-based fine-tuning to multi-modal Transformer-based vision-language models, enabling principled modulation of attention jointly in vision and text to offer excellent performance while better preserving model generalization. Furthermore, by rescaling semantic subspace vectors of pre-trained CLIP models via singular value modulation, CLIP-SVD explicitly reveals which pretrained feature directions are reused, suppressed, or amplified. This *subspace-level interpretability* offers a new lens into VLM adaptation mechanisms, allowing diagnosis of fine-grained adaptation dynamics, which is an aspect absent from prior works like CLIP-LoRA that introduce new degrees of freedom and thus obscure these observations.

Recent studies (Shuttleworth et al., 2025) on LLM finetuning reveal that LoRA and its variants could result in shifted singular vectors of the model parameter space (called intruder dimensions), potentially causing forgetting of past knowledge. In contrast, our CLIP-SVD leverages the rich semantic representation of VLM models by freezing the pretrained singular vectors $U$ and $V$ of the parameter space and tuning only the singular values $\Sigma$. This allows us to recalibrate the "importance" of task-relevant subspaces without distorting the original model geometry. While freezing $U$ and $V$ may potentially limit the expressivity of the parameter space, we find that this is both suitable and effective in the few-shot setting in our work, provided that CLIP-like models have been trained extensively with fine-grained representations. Unlike fully supervised scenarios common in LLM fine-tuning, our adaptation method assumes access to only a handful of labeled examples per class. In such cases, introducing large numbers of randomly initialized or fully tunable parameters is suboptimal. Since many downstream tasks lie in a low intrinsic dimension, and pretraining implicitly shapes these subspace vectors, fine-tuning via a low-dimensional subspace often suffices (Aghajanyan et al., 2021). Additionally, our interpretability analysis using TextSpan (Gandelsman et al., 2023) also shows that tuning only $\Sigma$ results in semantically meaningful shifts in attention.

In our experiments, few-shot VLM adaptation in the biomedical vision domain is notably more challenging than in the natural domain, likely due to factors like complex and unintuitive image features and ambiguous boundaries, as shown in previous works (Koleilat et al., 2025b). We show that CLIP-SVD *bridges this domain gap*, achieving strong performance in both biomedical and natural vision settings. In contrast, as shown in Tables 2 and 3, methods like CLIP-LoRA that were proposed for natural domain benchmarks do not generalize as well to biomedical tasks.

We adopted a natural language-based technique to understand the impact and insights of CLIP-SVD in model adaptation. For the related analyses in the biomedical domain, we *constructed a new corpus of biomedical image descriptions* by leveraging large language models. Together, these results constitute the first extensive empirical evaluation of SVF in the vision-language model setting, highlighting its strengths in both performance and interpretability. Although it is shown to facilitate the interpretation, further validation

is still required in broader applications, particularly with domain experts. We will investigate this in the near future.

## 6 Conclusion

In conclusion, we introduced CLIP-SVD, a novel parameter-efficient adaptation method for CLIP models that finetunes only the singular values of the weight matrices while preserving their pre-trained structure. Our approach enables effective few-shot learning with minimal computational overhead, achieving state-of-the-art results across both natural and biomedical domains. Through extensive ablation studies, we demonstrated the critical role of singular value adaptation in enhancing task-specific feature extraction. Further analyses of the Output-Value circuit with a natural-language-based approach revealed that adapting singular values steers attention heads toward more specialized roles, thus opening doors for further investigation of CLIP's characteristics in broader applications. To further promote transparency and reproducibility, we publicly release the full CLIP-SVD codebase, including the new biomedical corpus at https://github.com/HealthX-Lab/CLIP-SVD.

### Acknowledgments

We acknowledge the support of the Natural Sciences and Engineering Research Council of Canada (NSERC) and the Fonds de recherche du Québec – Nature et technologies (B2X-363874). We also thank Dr. Leila Kosseim for her valuable assistance in evaluating the completeness and correctness of the generated biomedical corpus.

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

# A    Dataset Details

Following Zhou et al. (2022b) and Koleilat et al. (2025b), we conducted extensive experiments on 11 natural and 10 biomedical classification benchmark datasets to evaluate the effectiveness of the proposed CLIP-SVD. The natural datasets include ImageNet (Deng et al., 2009), Caltech101 (Fei-Fei et al., 2004), OxfordPets (Parkhi et al., 2012), StanfordCars (Krause et al., 2013), Flowers102 (Nilsback & Zisserman, 2008), Food101 (Bossard et al., 2014), FGVCAircraft (Maji et al., 2013), SUN397 (Xiao et al., 2010), DTD (Cimpoi et al., 2014), EuroSAT (Helber et al., 2019), and UCF101 (Soomro et al., 2012). The biomedical datasets consist of CTKidney (Islam et al., 2022), DermaMNIST (Tschandl et al., 2018; Codella et al., 2019), Kvasir (Pogorelov et al., 2017), RETINA (Porwal et al., 2018; Köhler et al., 2013), LC25000 (Borkowski et al., 2019), CHMNIST (Kather et al., 2016), BTMRI (Nickparvar, 2021), OCTMNIST (Kermany et al., 2018), BUSI (Al-Dhabyani et al., 2020), COVID-QU-Ex (Tahir et al., 2021), and KneeXray (Chen, 2018). For distribution shift experiments, we also included ImageNetV2 (Recht et al., 2019), ImageNet-Sketch (Wang et al., 2019), ImageNet-A (Hendrycks et al., 2021b), and ImageNet-R (Hendrycks et al., 2021a). Dataset statistics are provided in Tables S3 and S4.

# B    Biomedical Corpus Statistics

Our generated biomedical corpus was reviewed and annotated by a human rater experienced in radiology and medical image analysis. Figure S1 illustrates the frequency distribution of the seven semantic categories used to annotate the biomedical corpus: **Shape**, **Texture**, **Contrast**, **Movement**, **Location**, **Object/Tissue Type**, and **Condition**. The corpus is dominated by appearance-related descriptors (Shape and Texture), reflecting the strong reliance on structural cues in biomedical image interpretation. Location and Object/Tissue Type attributes also occur frequently, consistent with the clinical need to specify anatomical context and involved tissues. Condition reflects the status of the tissue and underlying diseases. Overall, this distribution highlights the semantic emphasis of the corpus and provides insight into the types of visual reasoning most commonly required for biomedical image understanding.

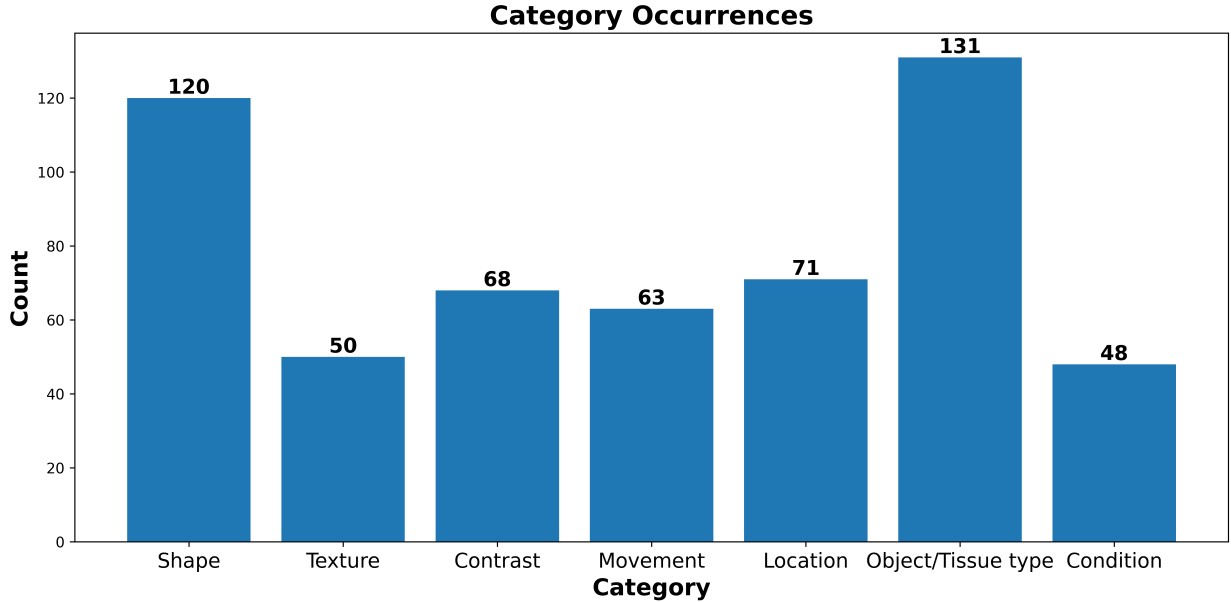

Figure S1: Distribution of semantic attribute categories annotated in the biomedical corpus

## C   Detailed Hyperparameters

The hyperparameters reported in Tables S1 and S2 across both natural and biomedical datasets were carefully tuned based on benchmark type and dataset characteristics. A consistent batch size (BS) of 32 was maintained throughout to ensure uniformity in training dynamics. For natural datasets, the learning rate (LR) was set to $5 \times 10^{-4}$ for few-shot settings and reduced to $6 \times 10^{-4}$ for base-to-novel evaluations. For the natural few-shot setting, we follow Zanella & Ben Ayed (2024) and set the number of iterations to $200 \times K$ (number of shots), whereas we use epochs for the base-to-novel setting, with the number of training epochs (EP) varying between 2 and 20 depending on the dataset's size. In biomedical datasets, LR values were more diverse, ranging from $0.5 \times 10^{-3}$ to $20 \times 10^{-3}$, reflecting the varying difficulty and modality of the medical tasks, while EP was generally higher (up to 200) to accommodate the typically smaller dataset sizes and slower convergence. Notably, BUSI (Al-Dhabyani et al., 2020) lacked base-to-novel evaluation due to the absence of appropriate class splits.

| Dataset | Benchmark | BS | LR ($10^{-4}$) | EP/IT |
|---|---|---|---|---|
| ImageNet | Few-shot | 32 | 5 | 200 |
| | Base-to-Novel | 32 | 6 | 10 |
| | Cross-dataset Transfer | 32 | 5 | 2 |
| | Domain Generalization | 32 | 5 | 2 |
| Caltech101 | Few-shot | 32 | 5 | 200 |
| | Base-to-Novel | 32 | 6 | 20 |
| DTD | Few-shot | 32 | 5 | 200 |
| | Base-to-Novel | 32 | 6 | 20 |
| EuroSAT | Few-shot | 32 | 5 | 200 |
| | Base-to-Novel | 32 | 6 | 20 |
| StanfordCars | Few-shot | 32 | 5 | 200 |
| | Base-to-Novel | 32 | 6 | 10 |
| Flowers102 | Few-shot | 32 | 5 | 200 |
| | Base-to-Novel | 32 | 6 | 10 |
| FGVCAircraft | Few-shot | 32 | 5 | 200 |
| | Base-to-Novel | 32 | 6 | 20 |
| SUN397 | Few-shot | 32 | 5 | 200 |
| | Base-to-Novel | 32 | 6 | 10 |
| OxfordPets | Few-shot | 32 | 5 | 200 |
| | Base-to-Novel | 32 | 6 | 10 |
| UCF101 | Few-shot | 32 | 5 | 200 |
| | Base-to-Novel | 32 | 6 | 10 |
| Food101 | Few-shot | 32 | 5 | 200 |
| | Base-to-Novel | 32 | 6 | 10 |

| Dataset | Benchmark | BS | LR ($10^{-3}$) | EP |
|---|---|---|---|---|
| BTMRI | Few-shot | 32 | 7 | 100 |
| | Base-to-Novel | 32 | 9 | 100 |
| BUSI | Few-shot | 32 | 2 | 100 |
| | Base-to-Novel | - | - | - |
| COVID-QU-Ex | Few-shot | 32 | 0.5 | 100 |
| | Base-to-Novel | 32 | 2 | 50 |
| CTKIDNEY | Few-shot | 32 | 1 | 200 |
| | Base-to-Novel | 32 | 10 | 200 |
| Kvasir | Few-shot | 32 | 5 | 100 |
| | Base-to-Novel | 32 | 3 | 60 |
| CHMNIST | Few-shot | 32 | 1 | 60 |
| | Base-to-Novel | 32 | 7 | 100 |
| LC25000 | Few-shot | 32 | 3 | 100 |
| | Base-to-Novel | 32 | 10 | 100 |
| RETINA | Few-shot | 32 | 7 | 100 |
| | Base-to-Novel | 32 | 20 | 60 |
| KneeXray | Few-shot | 32 | 8 | 100 |
| | Base-to-Novel | 32 | 2 | 60 |
| OCTMNIST | Few-shot | 32 | 10 | 100 |
| | Base-to-Novel | 32 | 20 | 100 |

Table S1: Hyperparameter values across natural datasets and benchmarks. (BS = Batch Size, LR = Learning Rate, EP = Epochs, IT = Iterations)

Table S2: Hyperparameter values across biomedical datasets and benchmarks. (BS = Batch Size, LR = Learning Rate, EP = Epochs)

## D   Domain Generalization

We evaluate the robustness of our method on out-of-distribution datasets. Similar to cross-dataset evaluation, we test our ImageNet-trained model directly on four other ImageNet datasets with different types of distribution shifts, including ImageNetV2 (Recht et al., 2019), ImageNet-Sketch (ImageNet-S) (Wang et al., 2019), ImageNet-A (Hendrycks et al., 2021b), and ImageNet-R (Hendrycks et al., 2021a). Table S5 presents the domain generalization results, where methods are trained on ImageNet and evaluated on datasets with various domain shifts (-V2, -S, -A, and -R). CLIP-SVD achieves the highest average accuracy of 62.55%, outperforming MaPLe (62.36%) and other methods. It also leads in two target domains, namely ImageNet-S

Table S3: **Summary of natural datasets:** Overview of the datasets used in the natural domain, including the number of classes, dataset splits (train, validation, test), and the corresponding hand-crafted text prompts used for classification.

| Dataset | Classes | Train | Val | Test | Hand-crafted prompt |
|---|---|---|---|---|---|
| ImageNet | 1,000 | 1.28M | N/A | 50,000 | "a photo of a [CLASS]." |
| Caltech101 | 100 | 4,128 | 1,649 | 2,465 | "a photo of a [CLASS]." |
| OxfordPets | 37 | 2,944 | 736 | 3,669 | "a photo of a [CLASS], a type of pet." |
| StanfordCars | 196 | 6,509 | 1,635 | 8,041 | "a photo of a [CLASS]." |
| Flowers | 102 | 4,093 | 1,633 | 2,463 | "a photo of a [CLASS], a type of flower." |
| Food101 | 101 | 50,500 | 20,200 | 30,300 | "a photo of [CLASS], a type of food." |
| FGVCAircraft | 100 | 3,334 | 3,333 | 3,333 | "a photo of a [CLASS], a type of aircraft." |
| SUN397 | 397 | 15,880 | 3,970 | 19,850 | "a photo of a [CLASS]." |
| DTD | 47 | 2,820 | 1,128 | 1,692 | "[CLASS] texture." |
| EuroSAT | 10 | 13,500 | 5,400 | 8,100 | "a centered satellite photo of [CLASS]." |
| UCF101 | 101 | 7,639 | 1,898 | 3,783 | "a photo of a person doing [CLASS]." |
| ImageNetV2 | 1,000 | N/A | N/A | 10,000 | "a photo of a [CLASS]." |
| ImageNet-Sketch | 1,000 | N/A | N/A | 50,889 | "a photo of a [CLASS]." |
| ImageNet-A | 1,000 | N/A | N/A | 50,889 | "a photo of a [CLASS]." |
| ImageNet-R | 1,000 | N/A | N/A | 50,889 | "a photo of a [CLASS]." |

Table S4: **Summary of biomedical datasets:** Overview of the datasets used in the biomedical domain, including the number of classes, dataset splits (train, validation, test), and the corresponding hand-crafted text prompts used for classification.

| Dataset | Classes | Train | Val | Test | Hand-crafted prompt |
|---|---|---|---|---|---|
| CTKIDNEY | 4 | 6,221 | 2,487 | 3,738 | "a photo of a [CLASS]." |
| Kvasir | 8 | 2,000 | 800 | 1,200 | "a photo of a [CLASS]." |
| RETINA | 4 | 2,108 | 841 | 1,268 | "a photo of a [CLASS]." |
| LC25000 | 5 | 12,500 | 5,000 | 7,500 | "a photo of a [CLASS]." |
| CHMNIST | 8 | 2,496 | 1,000 | 1,504 | "a photo of [CLASS]." |
| BTMRI | 4 | 2,854 | 1,141 | 1,717 | "a photo of a [CLASS]." |
| OCTMNIST | 4 | 97,477 | 10,832 | 1,000 | "a photo of a [CLASS]." |
| BUSI | 3 | 389 | 155 | 236 | "a photo of a [CLASS]." |
| COVID-Qu-Ex | 4 | 10,582 | 4,232 | 6,351 | "a chest xray of [CLASS]." |
| KneeXray | 5 | 5,778 | 826 | 1,656 | "a photo of a [CLASS]." |

(-S) (+0.47%) and ImageNet-V2 (-V2) (+0.28%), demonstrating its robustness to domain shifts. These results highlight the effectiveness of CLIP-SVD in generalizing across domains.

## E  Effect of Rank-based Singular Value Selection for CLIP-SVD

For CLIP-SVD, selective finetuning of the singular values could further reduce the computational requirement. With singular value decomposition, the singular values are naturally ranked in descending order with respect to their magnitude. In this experiment, we varied the number of non-zero singular values out of the full rank to be adapted according to their magnitude ordering, in order to investigate their impact on 4-shot model performance across both natural and biomedical domains. Specifically, we tested two configurations: one where only the top singular values (in descending order) are fine-tuned, and the other where only the bottom singular values (in ascending order) are adjusted. The results shown in Fig. S2 reveal that, in the

natural domain, both top and bottom configurations exhibit a steep initial accuracy increase starting from a small proportion of adjustable singular values, stabilizing near 78.18% as the ratio approaches full-rank, suggesting a limited sensitivity to the ranking of the selected singular values to be adapted. In contrast, the biomedical domain shows a more pronounced dependency on the ranking of the adjustable singular values. Here, finetuning the top singular values consistently outperforms the bottom configuration, reaching 68.02% at a full rank ratio, while the bottom approach lags behind by approximately 5% at the 0.5 ratio, highlighting the importance of prioritizing top singular values for specialized medical contexts.

Table S5: **Domain generalization:** Methods are trained on ImageNet using 16-shots and evaluated on datasets with domain shifts (ImageNet-V2, -S, -A, and -R) for classification accuracy (%).

| | Source | Target | | | | |
|---|---|---|---|---|---|---|
| | ImageNet | -V2 | -S | -A | -R | Avg. |
| CLIP | 66.73 | 60.83 | 46.15 | 47.77 | 73.96 | 59.09 |
| CoOp | 71.51 | 64.20 | 47.99 | 49.71 | 75.21 | 61.72 |
| Co-CoOp | 71.02 | 64.07 | 48.75 | 50.63 | 76.18 | 62.13 |
| CLIP-Adapter | 68.46 | 59.55 | 39.88 | 38.83 | 64.62 | 54.27 |
| TIP-Adapter | 53.81 | 45.69 | 29.21 | 36.04 | 55.26 | 44.00 |
| TIP-Adapter-F | 51.71 | 43.07 | 27.13 | 27.04 | 45.07 | 38.80 |
| TaskRes | 70.84 | 62.15 | 43.76 | 43.91 | 71.59 | 58.45 |
| KgCoOp | 71.20 | 64.10 | 48.97 | 50.69 | 76.70 | 62.33 |
| ProGrad | **72.24** | 64.73 | 47.61 | 49.39 | 74.58 | 61.71 |
| MaPLe | 70.72 | 64.07 | 49.15 | **50.90** | **76.98** | 62.36 |
| **CLIP-SVD** | 72.15 | **65.03** | **49.62** | 49.17 | 76.79 | **62.55** |

# F   Experiments with Other Backbones

In this experiment, we evaluated our proposed method, CLIP-SVD, using the alternative ViT-B/32 backbone in a few-shot learning setting for the natural domain. Table S6 presents the average classification accuracy (%) across 11 natural domain benchmarks derived from three randomly sampled support sets for each dataset. Our method consistently outperforms the state-of-the-art techniques across different few-shot settings. Notably, CLIP-SVD achieves the highest performance in all cases, surpassing the second-best method, Tip-Adapter-F, by a significant margin, with an improvement of approximately 1.2% at $K = 1$ and 1.4% at $K = 16$. These results highlight the strength of our singular value decomposition-based adaptation strategy in enhancing generalization under limited data conditions. Furthermore, the effectiveness of CLIP-SVD with the ViT-B/32 backbone demonstrates that our approach is adaptable to different model backbones, making it broadly applicable across various vision-language models.

Table S6: **Evaluation against state-of-the-art techniques for natural domain with ViT-B/32 backbone:** This table presents the average classification accuracy (%) obtained from 11 natural domain benchmarks derived from 3 sampled support sets for each dataset. The top-performing results are in bold, and the second-best are underlined.

| Method | $K = 1$ | $K = 2$ | $K = 4$ | $K = 8$ | $K = 16$ |
|---|---|---|---|---|---|
| Zero-shot CLIP (Radford et al., 2021) | | | 61.8 | | |
| CoOp (Zhou et al., 2022b) | 62.8 | 65.3 | 68.6 | 72.2 | 74.7 |
| CoCoOp (Zhou et al., 2022a) | 63.1 | 64.8 | 66.7 | 68.1 | 70.7 |
| ProGrad (Zhu et al., 2024a) | 64.7 | 67.1 | 69.8 | 73.2 | 75.9 |
| KgCoOp (Yao et al., 2023a) | 64.9 | 66.6 | 68.4 | 70.5 | 72.1 |
| MaPLe (Khattak et al., 2023a) | 61.5 | 65.2 | 68.7 | 71.6 | 74.1 |
| CLIP-Adapter (Gao et al., 2024) | 62.5 | 63.5 | 64.3 | 67.6 | 71.6 |
| Tip-Adapter-F (Zhang et al., 2021) | 66.5 | 69.2 | 71.6 | 74.1 | 77.1 |
| **CLIP-SVD (Ours)** | **67.7** | **71.0** | **73.4** | **75.8** | **78.5** |

Table S7: **Natural Domain Efficiency comparison** of different parameter-efficient tuning methods. We report trainable parameter counts, training, and inference time.

| Method | Trainable Params | Training Time (min) | Inference Time (s) |
|---|---|---|---|
| CoOp | 2.0K | 1.84 | 7.34 |
| CoCoOp | 35.4K | 2.25 | 18.52 |
| MaPLe | 3.5M | 2.95 | 7.40 |
| CLIP-Adapter | 131.1K | 4.93 | 7.48 |
| Tip-Adapter | 0 | – | 24.97 |
| TCP | 331.9K | 1.02 | 7.58 |
| CLIP-LoRA | 184.3K | 9.82 | 7.41 |
| CLIP-SVD (Ours) | 92.2K | 0.88 | 7.13 |

Table S8: **Biomedical Domain Efficiency comparison** of different parameter-efficient tuning methods. We report trainable parameter counts, training, and inference time.

| Method | Trainable Params | Training Time (min) | Inference Time (s) |
|---|---|---|---|
| CoOp | 2.0K | 1.19 | 7.20 |
| CoCoOp | 44.8K | 0.58 | 23.6 |
| MaPLe | 5.3M | 2.4 | 19.8 |
| CLIP-Adapter | 131.1K | 2.85 | 7.79 |
| Tip-Adapter | 0 | – | 23.52 |
| DCPL | 5.5M | 2.58 | 177.6 |
| BiomedCoOp | 3.1K | 1.76 | 7.20 |
| CLIP-LoRA | 221.2K | 10.25 | 7.58 |
| CLIP-SVD (Ours) | 110.6K | 1.78 | 6.76 |

## G  TextSpan Analysis Details

We analyze singular value shifts in the Output-Value (OV) circuit using `TextSpan` (Gandelsman et al., 2023), focusing on the last four layers (L8–L11) of CLIP and BiomedCLIP ViT-B/16 (*Note: layer and head indexing in these experiments begins at 0.*). The ViT-B/16 vision encoder employs 12 attention heads per MHSA block. For the natural image domain, we use ImageNet (Deng et al., 2009), while for the biomedical domain, we aggregate all relevant datasets (Islam et al., 2022; Pogorelov et al., 2017; Porwal et al., 2018; Köhler et al., 2013; Borkowski et al., 2019; Kather et al., 2016; Nickparvar, 2021; Kermany et al., 2018; Al-Dhabyani et al., 2020; Tahir et al., 2021; Chen, 2018) into a single comprehensive dataset. Images from both domains are fed into their corresponding vision encoder network. For the natural domain, we utilize the text corpus from Gandelsman et al. (2023), which consists of approximately 3,000 GPT-3.5-generated general image descriptions. In contrast, no large-scale text corpus exists for biomedical images. To address this, we generated a new corpus comprising 300

Table S9: Relevance scores (0–5) for the top 3 Attention Heads with the highest normalized changes after CLIP adaptation. "**L**" denotes layer and "**H**" denotes attention head.

| EuroSAT (Satellite Images) | DTD (Texture Images) |
|---|---|
| (**L10**.**H0**): 5.00 | (**L8**.**H6**): 2.00 |
| (**L10**.**H10**): 4.67 | (**L10**.**H2**): 5.00 |
| (**L11**.**H0**): 3.33 | (**L9**.**H4**): 4.00 |
| SUN397 (Scene Understanding) | UCF101 (Action Recognition) |
| (**L11**.**H0**): 5.00 | (**L10**.**H5**): 1.67 |
| (**L11**.**H2**): 3.00 | (**L10**.**H6**): 4.33 |
| (**L11**.**H3**): 5.00 | (**L10**.**H1**): 4.33 |
| BUSI (Breast Ultrasound) | BTMRI (Brain MRI) |
| (**L11**.**H8**): 4.67 | (**L8**.**H9**): 5.00 |
| (**L8**.**H6**): 3.33 | (**L8**.**H0**): 2.33 |
| (**L8**.**H3**): 5.00 | (**L9**.**H5**): 4.00 |
| COVID-QU-Ex (Chest X-ray) | CTKIDNEY (Kidney CT) |
| (**L11**.**H0**): 3.33 | (**L8**.**H6**): 4.00 |
| (**L9**.**H1**): 4.67 | (**L8**.**H3**): 4.67 |
| (**L11**.**H3**): 3.00 | (**L8**.**H10**): 4.00 |

general medical image descriptions and relevant terminologies using GPT-4 (Achiam et al., 2023). We use the following prompt to query GPT-4: `"Generate 300 distinct descriptive prompts for medical images that capture specific visual features commonly found in medical scans, such as contrast, shape, color, location, and texture."` Using `TextSpan`, we extract the top three text descriptions from the corpus that best characterize the role of each attention head in each of the final four layers. Based on these top-3 descriptions, GPT-4 is used to assign a concise and descriptive title to each head's function: `"Assign a concise and descriptive title that best captures the primary function represented by these descriptions."`. The identified roles and associated top-3 descriptions for each head in layers L8–L11 for the natural and biomedical domains are detailed in Tables S13 and S14, respectively.

To further validate the semantic relevance of the layer/head rankings, we conduct an additional evaluation using an LLM-as-a-judge framework (Pavlovic & Poesio, 2024; Schoenegger et al., 2024; Kim et al., 2023). Specifically, three independent large language models (GPT-5 (OpenAI, 2025), Qwen3 (Yang et al., 2025), and Gemini 3 (Google, 2025)) are prompted to assign a relevance score from 1 to 5 to each head description,

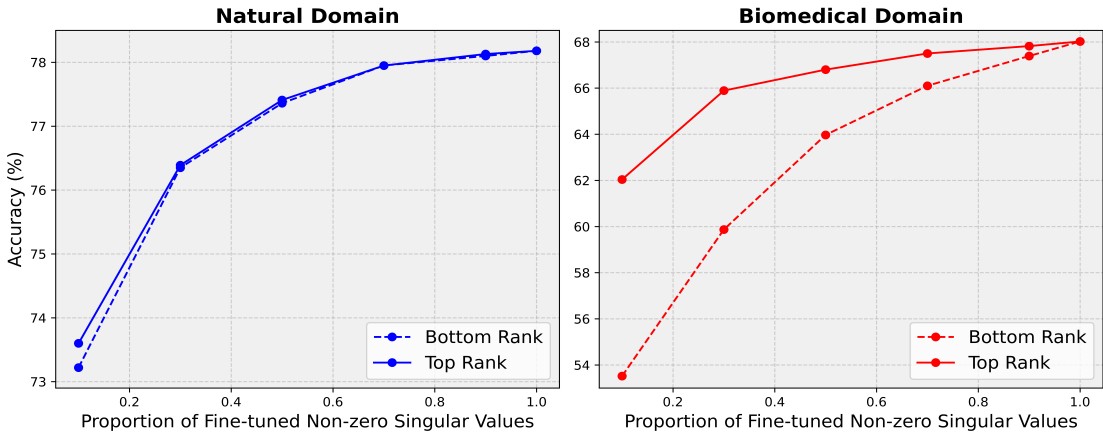

Figure S2: Accuracy comparison for models finetuned with varying rank ratios in the natural (left) and biomedical (right) domains

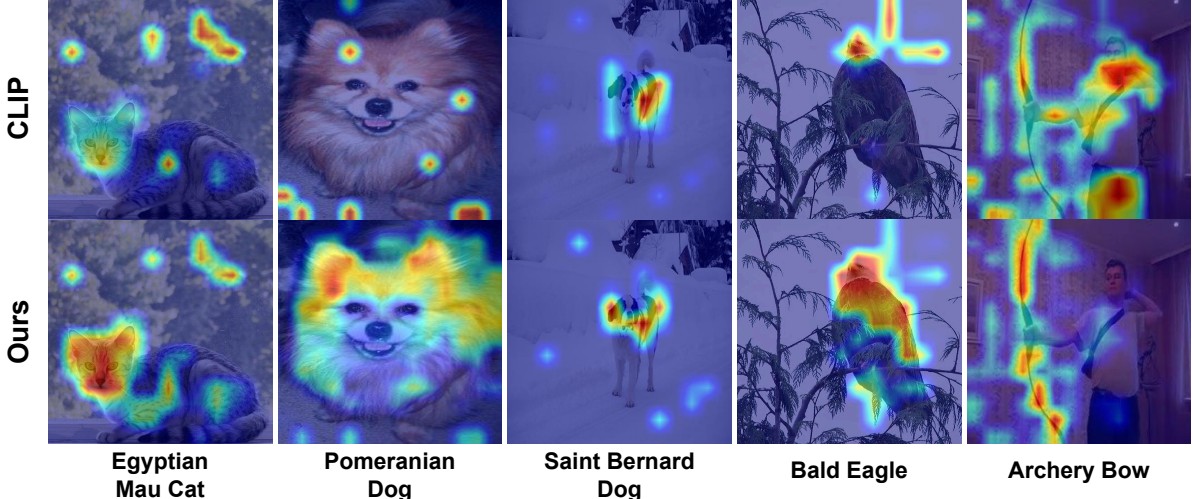

Figure S3: Saliency map comparison between the pre-trained CLIP model and the CLIP-SVD fine-tuned model. The adapted model shows more focused and semantically aligned attention, highlighting improved localization of relevant image regions.

where 1 indicates a limited meaningful connection, and 5 indicates strong alignment with the dataset's domain-specific visual characteristics. As shown in Table S9, the three scores are averaged to produce a final relevance grade for each head. Across all evaluated heads and layers, the mean relevance score is **3.97**, indicating that the TextSpan-derived descriptions are highly aligned with the visual features most important for interpreting images in each domain. This experiment provides an external, model-agnostic measure of the validity of the extracted semantic roles.

# H    t-SNE Visualization

We used t-SNE (van der Maaten & Hinton, 2008) to visualize the image embeddings generated by the fine-tuned model using our CLIP-SVD and compared them with the embeddings from the pre-trained CLIP and BiomedCLIP models. As shown in Figure S4, the embeddings produced by CLIP-SVD exhibit significantly better clustering, indicating that our method enables the model to generate more distinct and well-separated feature representations. This experiment was conducted on the EuroSAT, OxfordPets, CTKidney,

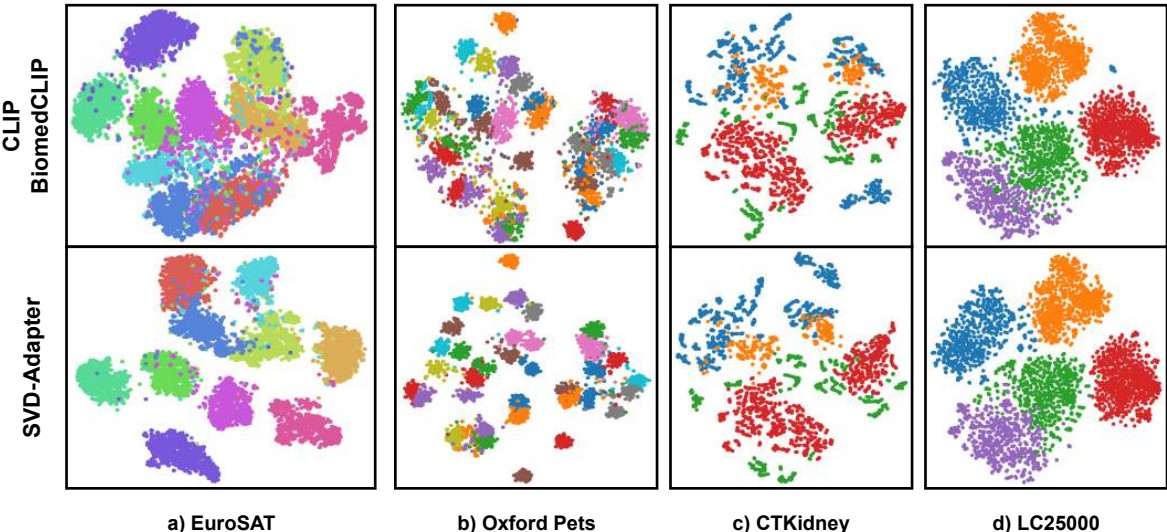

Figure S4: **t-SNE visualization of image embeddings:** Comparison of embeddings produced by the pre-trained CLIP/BiomedCLIP model and the fine-tuned CLIP-SVD on (a) EuroSAT, (b) Oxford Pets, (c) CTKidney, and (d) LC25000. CLIP-SVD generates more compact and well-separated clusters, indicating improved feature extraction and task-specific alignment.

and LC25000 datasets, where the improved clustering patterns highlight the enhanced feature extraction capabilities of our approach.

## I  Segmentation Performance

We evaluated the segmentation performance of our fine-tuned model using the TextSpan algorithm (Gandelsman et al., 2023) on the ImageNet-Segmentation dataset (Guillaumin et al., 2014), a curated subset of 4,276 ImageNet validation images with pixel-level annotations, and compared the results against those of the original pre-trained CLIP model.

To perform segmentation, we follow the approach introduced in the TextSpan framework (Gandelsman et al., 2023), which builds on a fine-grained decomposition of CLIP's image representation. Specifically, we use the fact that the CLIP image encoder's output can be expressed as a sum over attention head contributions across spatial positions. Each image patch contributes a vector in the shared image-text embedding space. Given a text prompt corresponding to the object class (i.e. `"an image of a [CLASS]"`), we compute a heatmap by measuring the similarity between each patch's contribution and the CLIP

| Model | Pixel Acc. | mIoU | mAP |
|---|---|---|---|
| CLIP | 77.24 | 57.73 | 82.62 |
| CLIP-SVD (Ours) | **77.55** | **58.27** | **82.82** |
| Δ | +0.31 | +0.54 | +0.20 |

Table S10: Segmentation Performance in terms of pixel accuracy, mean IoU, and mean average precision (%).

embedding of the text description. These similarity scores are then aggregated into a spatial map that highlights the regions most semantically aligned with the prompt. We also used gScoreCAM (Chen et al., 2022a) to visualize saliency maps for both the pre-trained CLIP model and the version fine-tuned with CLIP-SVD. Table S10 shows the results in terms of pixel accuracy, mean intersection over union (mIoU), and mean average precision (mAP). Our method achieves a performance boost across all metrics, with a gain of +0.31% in pixel accuracy, +0.54% in mIoU, and +0.20% in mAP. These improvements demonstrate that CLIP-SVD enhances the model's ability to localize and segment fine-grained regions in complex natural images, leading to more accurate and consistent segmentation results. This boost in segmentation performance stems from improved feature extraction and attention alignment through singular value adaptation as indicated in Section 4.6. Fine-tuning the singular values helps the model's attention heads capture task-relevant features, such as geographic cues in EuroSAT and color-related patterns in Oxford Pets, as shown in Figure S3. This

enables better foreground-background differentiation and more precise segmentation boundaries, explaining the observed performance gains.

Table S11: **Trainable parameter counts for CLIP and BiomedCLIP encoders under the full-rank SVD formulation.**

| Model / Encoder | Hidden Size $D$ | MHSA Params ($4D$) | MLP Params ($2D$) | Total per Layer ($6D$) | Total per Encoder (12 Layers = $72D$) |
|---|---|---|---|---|---|
| **CLIP Text Encoder** | 512 | 2,048 | 1,024 | 3,072 | 36,864 |
| **CLIP ViT-B/16 Vision Encoder** | 768 | 3,072 | 1,536 | 4,608 | 55,296 |
| **CLIP Total (Text + Vision)** | — | — | — | — | **92,160** |
| **BiomedCLIP Text (PubMedBERT)** | 768 | 3,072 | 1,536 | 4,608 | 55,296 |
| **BiomedCLIP Vision (ViT-B)** | 768 | 3,072 | 1,536 | 4,608 | 55,296 |
| **BiomedCLIP Total (Text + Vision)** | — | — | — | — | **110,592** |

Table S12: **Pre-processing SVD Computational Cost and Memory Usage**

| Method | Time per Block (ms) | Total Time (s) | Peak GPU Memory (GB) |
|---|---|---|---|
| CLIP | 84.46 | 12.16 | 1.83 |
| BiomedCLIP | 103.58 | 15.23 | 2.46 |

## J  Computational Cost

We present detailed results in Tables S7 and S8, comparing various PEFT methods in terms of total trainable parameters, training time, and inference time on the DTD and RETINA datasets, respectively. This analysis highlights the performance–efficiency trade-off across methods. All experiments were conducted on a single NVIDIA A100 GPU with 40GB of RAM, using a batch size of 8 for inference for consistency and fairness. The results demonstrate that CLIP-SVD strikes a favorable balance: it requires relatively few trainable parameters, converges more quickly, and introduces no inference overhead, while achieving high accuracy in few-shot settings. Moreover, CLIP-SVD avoids reliance on external modules or architectural modifications, simplifying deployment and improving efficiency, particularly in low-resource or latency-sensitive scenarios. Table S11 reports the detailed parameter counts of the CLIP and BiomedCLIP backbones under the full-rank SVD formulation. These include hidden dimension sizes, MHSA and MLP parameter counts, and the total number of trainable parameters per layer and per encoder. As shown, CLIP encoders contain a total of 92,160 SVD parameters across text and vision backbones, while BiomedCLIP contains 110,592 parameters due to its larger PubMedBERT-based text encoder. From a computational perspective, CLIP-SVD introduces an explicit SVD decomposition step that differs from the random initialization used in LoRA-based adaptations. In practice, this decomposition is performed once at the model level for the pretrained CLIP or BiomedCLIP backbone and does not depend on the downstream task, dataset, or adaptation configuration. The resulting SVD-parameterized zero-shot model can then be stored and reused across all few-shot adaptation experiments, without requiring any additional SVD computation. As reported in Table S12, the one-time preprocessing cost is modest (12.16s for CLIP and 15.23s for BiomedCLIP) with moderate GPU memory usage. After this preprocessing stage, training and inference proceed without additional computational or memory overhead, since adaptation is restricted to updating the singular values. While this initial cost exceeds that of random initialization in LoRA, it is incurred once and shared across tasks, and therefore does not materially impact the efficiency of CLIP-SVD in practical multi-task adaptation settings.

## K  Additional Per-dataset Results

We provide the complete per-dataset few-shot results for both the natural and biomedical domains to offer a detailed evaluation of our method's performance. The natural and biomedical domain results are summarized in Tables S15 and S16, respectively, which report the classification accuracy across different few-shot settings for each dataset. On the other hand, we offer the per-dataset base-to-novel generalization results in Table S17 and S18. This detailed breakdown highlights the consistent improvement achieved by CLIP-SVD over state-of-the-art methods, demonstrating its ability to generalize effectively across diverse datasets and domains.

Table S13: Natural Domain Analysis of the attention heads of each layer with the top 3 descriptions returned by TextSpan(Gandelsman et al., 2023). Here, "**L**" denotes layer while "**H**" denotes attention head.

| L8.H0: Foundational Shapes & Silhouettes | L8.H1: Motion & Directional Cues |
|---|---|
| Rural windmill silhouette
A plank
Intricate clock mechanism | Dynamic sports action shot
Tranquil boating on a lake
Lively city parade |
| L8.H2: Contrast & Light-Dark Regions | L8.H3: Geometric Patterns & Grids |
| Photograph with the artistic style of fisheye lens
Psychedelic color swirls
Sunlit meadow path | Artwork featuring Morse code typography
Cubist still life painting
Miniature diorama photography |
| L8.H4: Object Boundaries & Contours | L8.H5: Repeated Structures & Textures |
| Cinematic portrait with dramatic lighting
A whirlpool
Image with harlequin patterns | Plaid pattern
Reflective surfaces
Image of a scooter |
| L8.H6: Refined Textural Details of Everyday Objects | L8.H7: Positional Layout & Spatial Balance |
| Collage of textures
Close-up of a textured bark
Mesmerizing kinetic sculpture | Cozy cabin interior
Minimalist white backdrop
Rolling vineyard landscapes |
| L8.H8: Edge Detection & Detail Refinement | L8.H9: Primitive Forms & Visual Anchors |
| A scalene quadrilateral
Minimalist architectural photography
Detailed charcoal sketch | Dappled sunlight
A cloverleaf
Ocean sunset silhouette |
| L8.H10: Coarse-to-Fine Visual Transitions | L8.H11: Structural Symmetry & Alignment |
| Intricate wood carving
Contemplative monochrome portrait
Aerial view of a marketplace | Birds-eye view
Detailed illustration of a futuristic brain-computer interface
Tranquil garden pathway |
| L9.H0: Scene Composition & Perspective | L9.H1: Depth Perception & Visual Planes |
| Aerial view of a marketplace
Glowing neon cityscape
Playful siblings | A zoomed out photo
Classic black and white cityscape
Antique historical artifact |
| L9.H2: Interaction Between Objects | L9.H3: Material Surfaces & Reflection |
| An image of a Gymnast
Emotional and heartfelt human connection
Detailed illustration of a futuristic brain-computer interface | Aerial view of a teeming rainforest
Inviting reading nook
Dynamic and high-energy dance competition |
| L9.H4: Natural Landscapes & Textures | L9.H5: Architectural Layouts & Forms |
| Aerial view of a vineyard
Enchanting forest nymph aesthetic
Delicate and intricate lace patterns | Photograph showcasing laughter
Minimalist architectural photography
Urban labyrinth |
| L9.H6: Facial Features & Gaze Cues | L9.H7: Semantic Grouping of Elements |
| Detailed charcoal sketch
Intense water sports moment
Photograph with the artistic style of light trails | Urban rooftop panorama
Cozy cabin interior
Photo taken in the Japanese tea gardens |
| L9.H8: Soft Lighting & Shadow Play | L9.H9: Organic Flow & Movement |
| Impressionist portrait painting
Intriguing and enigmatic passageway
Ephemeral soap bubble pattern | Photograph with abstract geometric overlay
Artwork featuring Morse code typography
Abstract oil painting |
| L9.H10: Conceptual Boundaries & Themes | L9.H11: Temporal Sequences & Visual Narratives |
| Miniature diorama photography
emotional candid moment
Artwork featuring Morse code typography | Colorful hot air balloons
Picture taken in Rwanda
Joyful family picnic scene |

Table S13 (continued): Natural Domain Analysis of the attention heads of each layer with the top 3 descriptions returned by TextSpan (Gandelsman et al., 2023). Here, "**L**" denotes layer while "**H**" denotes attention head.

| L10.H0: Aerial Landscapes & Environments | L10.H1: Human Experiences & Objects in Action |
|---|---|
| Aerial view of an agricultural field
Image taken in the Namibian desert
Picture taken in the Brazilian rainforest | Hands in an embrace
Dynamic and high-energy music concert
Emergency Medical Technician at work |
| **L10.H2: Cultural & Textural Scenes** | **L10.H3: Colorful Festivities & Patterns** |
| Picture taken in the Spanish Flamenco festivals
Ornate wood carving
Image captured in the Peruvian Andes | Lively and colorful parade
Tie-dye design
Soft pastel tones |
| **L10.H4: Cozy Indoors & Historical Locations** | **L10.H5: Action, Emotion & Faces** |
| Photograph taken in a retro diner
Picture taken in the Scottish castles
Cozy living room ambiance | Dynamic Action
Energetic children
Playful winking facial expression |
| **L10.H6: Organic Forms & Flow** | **L10.H7: Geographic & Cultural Diversity** |
| Graceful wings in motion
Curved organic designs
Ethereal double exposure photography | Picture taken in Pakistan
Bustling cultural market
Image snapped in the Swiss chocolate factories |
| **L10.H8: Color & Design Aesthetics** | **L10.H9: Sketches, Illustrations & Concepts** |
| Photograph with a yellow color palette
Photo taken in the Brazilian samba parade
Colorful hot air balloons | Collage of vintage magazine clippings
Detailed charcoal sketch
Impressionist-style digital painting |
| **L10.H10: Mood, Atmosphere & Highlights** | **L10.H11: Human-Centered Realism & Objects** |
| Image with holographic retro synthwave aesthetics
Serene lakeside cabin
Whirlpool of brimstone | Urban labyrinth
Image of a police car
Intricate gemstone arrangement |
| **L11.H0: Semantic Layout & Spatial Context** | **L11.H1: Human Relationships & Portraiture** |
| Mysterious day scene
Urban rooftop panorama
A zoomed out photo | Images of people and emotional interaction
Family portraits and group dynamics
Visuals of children, couples, and professionals |
| **L11.H2: Numbers, Symbols & Temporal Cues** | **L11.H3: Lifestyle & Tranquil Activities** |
| Digits and symbolic representations in images
Temporal cues like clocks, sunsets, motion blur
Confetti, reflections, and expressive timing | Calm urban and rural lifestyle scenes
Tranquil locations: diners, libraries, markets
Cozy homes, beach sunsets, fairgrounds |
| **L11.H4: Visual Symbols & Cultural Motifs** | **L11.H5: Rare Words & Abstract Concepts** |
| Cultural symbols and calligraphy art
Caricatures, retro TV test patterns, mystical motifs
Architectural and folkloric aesthetics | Abstract words (quasar, zephyr), letter imagery
Rare geographic terms, numerals, and geometry
Conceptual elements and visual abstractions |
| **L11.H6: Global Geographic Locations** | **L11.H7: Color Themes & Tonal Palettes** |
| Images tied to global cities and landmarks
Natural and urban landscapes across continents
Famous locations: Machu Picchu, Santorini, NYC | Dominant color schemes (red, blue, gray)
Pastel, sepia tones, artistic overlays
Color-coded compositions and political art |
| **L11.H8: Weather & Natural Forces** | **L11.H9: Everyday Objects & Refined Details** |
| Storms, fog, lightning, and natural disasters
Atmospheric effects and visual phenomena
Wildlife amidst dynamic weather elements | Common items like scarves, bowls, wallets
Mechanisms, furniture, modern artifacts
Textures: wood grain, bubble, fabric |
| **L11.H10: Natural Flora, Landscapes & Tranquility** | **L11.H11: Living Creatures & Natural Forms** |
| Floral and plant life in serene settings
Natural scenes: cherry blossoms, forests
Geographic tranquility and biodiversity | Wide range of animals: dogs, elephants, fish
Insects, feathers, natural surface textures
Shells, fur, fruit, snowflakes |

Table S14: Biomedical Domain Analysis of the attention heads of each layer with the top 3 descriptions returned by TextSpan (Gandelsman et al., 2023). Here, "**L**" denotes layer while "**H**" denotes attention head.

| L8.H0: Focal Markers & Shape Cues | L8.H1: Peripheral Edges & Enhancement Rings |
|---|---|
| Target signs and air bronchograms
Geographic zones of tissue involvement
Microcalcifications and pseudocapsule patterns | Ring-enhancing lesions and peripheral edema
Pseudocapsules and symmetric deviations
Finger-in-glove signs and disruption of growth plates |
| **L8.H2: Lobe Collapse & Serpentine Patterns** | **L8.H3: Radiologic Artifacts & Diffuse Shapes** |
| Collapsed lobes and reverse halo signs
Serpiginous lesions and high signal regions
Growth plate disruptions and cortical thinning | Comet-tail artifacts and nodular regions
Fluid levels suggesting benignity
Asymmetries and contrast-enhanced zones |
| **L8.H4: Flow Voids & Vascular Boundaries** | **L8.H5: Textural Shifts & Patchy Regions** |
| An area with decreased perfusion
A vascular displacement
A vascular structure with sharp borders | Mosaic attenuation and necrotic centers
Patchy regions with blurry margins
Air bronchograms and target-like patterns |
| **L8.H6: Contour Irregularity & Internal Spread** | **L8.H7: Symmetry Deviation & Dense Regions** |
| An anatomical displacement
A spiculated margin
A zone of tissue infiltration | Mass effect and asymmetry in dense zones
Patchy consolidations and star-shaped opacities
Air bronchograms and peri-lesional edema |
| **L8.H8: Clustered Signals & Midline Shifts** | **L8.H9: Scattered Highlights & Artifactual Spots** |
| Midline shifts with hyperintense clustering
Target signs and septated fluid collections
Cortical thinning and pseudocapsule formations | An area with decreased perfusion in the left hemisphere
A bright spot artifact in a clustered pattern
A contrast-enhanced region on axial view |
| **L8.H10: Diffuse Zones & Overlapping Shapes** | **L8.H11: Dense Regions & Local Signal Buildup** |
| Donut and target signs in scattered areas
Diffuse abnormalities and cross-compartment lesions
Hypo- and hyperintense signals across lobes | Zones of infiltration with surrounding edema
Septated collections and mass effects
Contrast enhancement and vascular structures |
| **L9.H0: Symmetry Shifts & Linear Features** | **L9.H1: Ring-Like Structures & Localized Spread** |
| A deviation from expected symmetry in the left hemisphere
A serpiginous lesion in the left hemisphere
A well-defined border suggesting benignity | A central necrosis in a scattered distribution
A crescent-shaped fluid collection in a scattered distribution
A deviation from anatomical landmarks consistent with inflammation |
| **L9.H2: Irregular Densities & Textured Borders** | **L9.H3: Small Patterned Anomalies** |
| A beam-hardening artifact
A collapsed lung lobe without enhancement
A peripherally enhancing collection with sharp borders | A tree-in-bud pattern without enhancement
A cavitary lesion without enhancement
A high signal-to-noise ratio with sharp borders |
| **L9.H4: Defined Edges & Subpleural Zones** | **L9.H5: Streaks, Texture, & Soft Borders** |
| A subpleural nodule
A central necrosis in a scattered distribution
A lesion crossing compartments | A zone of tissue infiltration with surrounding edema
A tram-track sign in the left hemisphere
A striated muscle texture |
| **L9.H6: Sharp Edges & Tissue Mismatch** | **L9.H7: Thickened Zones & Star-Like Forms** |
| An anatomical displacement
An irregular border with surrounding edema
A streak artifact | A region of tissue thickening
A star-shaped opacity in the upper lobe
A contrast-enhanced region |
| **L9.H8: Homogeneous Textures & Clustered Lines** | **L9.H9: Contour Disruptions & Shape Markers** |
| A lesion with fluid level
A homogeneous texture
A disruption of normal anatomy | A cystic lesion suggesting benignity
A reverse halo sign
A triangular-shaped defect |
| **L9.H10: Circumscribed Lesions & Flow Paths** | **L9.H11: Signal Artifacts & Internal Septations** |
| A well-circumscribed lesion
A cluster of microcalcifications suggesting malignancy
A serpiginous lesion in the left hemisphere | A comet-tail artifact
A lesion with internal septations without enhancement
A motion artifact |

Table S14 (continued): Biomedical Domain Analysis of the attention heads of each layer with the top 3 descriptions returned by TextSpan (Gandelsman et al., 2023). Here, "**L**" denotes layer while "**H**" denotes attention head.

| L10.H0: Low-Contrast Regions & Collapse Patterns | L10.H1: Localized Irregularities & Shadowing |
|---|---|
| A collapsed lung lobe
A vascular structure with sharp borders
A focal area of calcification in the left hemisphere | A deviation from anatomical landmarks
A lytic lesion consistent with inflammation
A finger-in-glove sign in the left hemisphere |
| L10.H2: Contrast Rings & Interface Blur | L10.H3: Multi-Compartment Spread & Fine Edges |
| An air bronchogram suggesting malignancy
A blurring of tissue interfaces
A peripherally enhancing collection consistent with infection | A lesion crossing compartments in the left hemisphere
A miliary pattern
A vascular structure with sharp borders |
| L10.H4: Fused Regions & Curved Forms | L10.H5: Sharp Transitions & Highlighted Foci |
| A finger-in-glove sign in the left hemisphere
A region with cortical thinning with surrounding edema
A lesion with fluid level suggesting benignity | A streak artifact
A targetoid lesion
A cluster of microcalcifications suggesting malignancy |
| L10.H6: Circular Opacities & Clean Boundaries | L10.H7: Textured Rings & Septated Centers |
| A circular opacity
A septated fluid collection
A clean fluid interface | A central necrosis
A lesion with calcified rim
A reverse halo sign |
| L10.H8: Artifact-Like Patterns & Symmetry Shift | L10.H9: Smooth Outlines & Diffuse Flow |
| A midline shift in a clustered pattern
A beam-hardening artifact
A disruption of growth plate suggesting benignity | A serpiginous lesion in the left hemisphere
A stenotic segment in the upper lobe
A lesion with irregular internal architecture |
| L10.H10: Star Patterns & Linear Disruptions | L10.H11: Texture Aggregates & Uniform Zones |
| A star-shaped opacity in the upper lobe
A shifting fluid level
A motion artifact suggesting benignity | A lytic lesion on axial view
A honeycomb pattern
A sunburst pattern suggesting benignity |
| L11.H0: Signal Voids & Midline Shifts | L11.H1: Blurred Regions & Artifact Shapes |
| A collapsed lung lobe
A low signal-to-noise ratio in the upper lobe
A lesion crossing compartments | A solid-cystic component suggesting malignancy
A finger-in-glove sign in a scattered distribution
A double-density sign suggesting benignity |
| L11.H2: Peripheral Contrast & Mosaic Attenuation | L11.H3: Cross-Lobe Flow & Density Buildup |
| A mosaic attenuation
A geographic area of involvement in the upper lobe
A peripherally enhancing collection consistent with infection | A cavitary lesion without enhancement
A lesion crossing compartments
A shifting fluid level |
| L11.H4: Sharp Fluid Interfaces & Contrast Pockets | L11.H5: Ringed Zones & Pattern Complexity |
| A tram-track sign on axial view
A clean fluid interface
A central necrosis with sharp borders | A targetoid lesion
A fluid-fluid level suggesting malignancy
A mosaic attenuation |
| L11.H6: Rounded Opacities & Midline Distortions | L11.H7: Compact Signal Shifts & Grid-Like Forms |
| A circular opacity
A solid-cystic component suggesting malignancy
A crescent-shaped fluid collection | A star-shaped opacity in the upper lobe
A widened mediastinum
A spiculated margin |
| L11.H8: Converging Edges & Cluster Markers | L11.H9: Scattered Intensity & Symmetry Cues |
| A low-contrast region in a clustered pattern
A double-density sign suggesting benignity
A solid-cystic component suggesting malignancy | A midline shift
A disruption of normal anatomy
A clean fluid interface |
| L11.H10: Rimmed Signals & Flow Distributions | L11.H11: Sharp Vascular Lines & Compact Zones |
| A peripherally enhancing collection consistent with infection
A change in signal intensity suggesting malignancy
A beam-hardening artifact | A vascular structure with sharp borders
A cluster of microcalcifications suggesting malignancy
A homogeneous texture |

Table S15: **Natural per-dataset performance** comparison of with various methods in few-shot setting in terms of classification accuracy (%).

| Dataset | Method | $K = 1$ | $K = 2$ | $K = 4$ | $K = 8$ | $K = 16$ |
|---|---|---|---|---|---|---|
| ImageNet | CLIP | | | 66.72 | | |
| | CLIP-Adapter | 67.93 | 68.60 | 69.56 | 70.20 | 71.60 |
| | Tip-Adapter | 68.80 | 68.90 | 69.68 | 70.01 | 70.36 |
| | Tip-Adapter-F | 67.43 | 68.60 | 69.86 | 71.40 | 73.10 |
| | Linear Probing | 31.30 | 43.83 | 54.20 | 61.93 | 67.27 |
| | LP++ | 69.29 | 69.63 | 70.80 | 72.10 | 72.95 |
| | CoOp | 65.77 | 68.17 | 69.38 | 70.83 | 71.51 |
| | CoCoOp | 69.51 | 69.84 | 70.55 | 70.77 | 71.02 |
| | KgCoOp | 69.03 | 69.63 | 70.19 | 70.23 | 71.2 |
| | ProGrad | 64.33 | 66.12 | 70.21 | 71.10 | 72.68 |
| | MaPLe | 62.67 | 65.10 | 67.70 | 70.30 | 72.33 |
| | CLIP-SVD (Ours) | 70.03 | 70.68 | 71.46 | 72.36 | 73.25 |
| Caltech101 | CLIP | | | 93.31 | | |
| | CLIP-Adapter | 93.30 | 93.67 | 94.00 | 94.27 | 94.57 |
| | Tip-Adapter | 93.08 | 93.43 | 94.19 | 94.21 | 94.50 |
| | Tip-Adapter-F | 93.56 | 94.22 | 95.06 | 95.20 | 95.79 |
| | Linear Probing | 80.08 | 87.00 | 92.27 | 93.85 | 95.35 |
| | LP++ | 92.85 | 94.14 | 94.87 | 95.44 | 95.94 |
| | CoOp | 92.37 | 92.83 | 94.44 | 94.10 | 95.50 |
| | CoCoOp | 93.80 | 94.92 | 94.98 | 95.11 | 95.19 |
| | KgCoOp | 94.13 | 94.20 | 94.65 | 94.97 | 95.03 |
| | ProGrad | 90.96 | 93.21 | 94.93 | 94.92 | 95.80 |
| | MaPLe | 92.57 | 93.97 | 94.43 | 95.20 | 96.00 |
| | CLIP-SVD (Ours) | 93.91 | 94.44 | 95.12 | 95.85 | 96.17 |
| DTD | CLIP | | | 44.09 | | |
| | CLIP-Adapter | 45.20 | 47.87 | 53.83 | 66.40 | 71.17 |
| | Tip-Adapter | 50.24 | 52.44 | 57.64 | 61.92 | 65.68 |
| | Tip-Adapter-F | 53.25 | 56.32 | 62.09 | 67.28 | 72.05 |
| | Linear Probing | 36.03 | 46.02 | 56.01 | 63.71 | 69.82 |
| | LP++ | 52.34 | 56.05 | 62.17 | 67.30 | 71.41 |
| | CoOp | 49.00 | 51.70 | 58.57 | 64.70 | 68.63 |
| | CoCoOp | 48.51 | 52.02 | 54.79 | 58.92 | 63.78 |
| | KgCoOp | 52.50 | 55.73 | 58.31 | 65.87 | 69.37 |
| | ProGrad | 52.79 | 54.35 | 57.72 | 62.13 | 65.92 |
| | MaPLe | 52.13 | 55.50 | 61.00 | 66.50 | 71.33 |
| | CLIP-SVD (Ours) | 56.05 | 60.60 | 64.83 | 68.28 | 72.42 |
| EuroSAT | CLIP | | | 48.37 | | |
| | CLIP-Adapter | 61.70 | 66.07 | 66.80 | 73.23 | 81.87 |
| | Tip-Adapter | 62.46 | 64.65 | 70.39 | 71.57 | 78.44 |
| | Tip-Adapter-F | 63.95 | 70.38 | 76.43 | 81.11 | 87.46 |
| | Linear Probing | 56.36 | 59.56 | 71.90 | 77.93 | 85.45 |
| | LP++ | 65.02 | 67.99 | 74.00 | 76.32 | 84.63 |
| | CoOp | 54.80 | 61.20 | 68.62 | 75.53 | 83.60 |
| | CoCoOp | 55.71 | 46.24 | 63.83 | 68.26 | 73.82 |
| | KgCoOp | 60.83 | 68.97 | 71.06 | 72.37 | 74.93 |
| | ProGrad | 55.10 | 66.19 | 70.84 | 79.22 | 84.38 |
| | MaPLe | 71.80 | 78.30 | 84.50 | 87.73 | 92.33 |
| | CLIP-SVD (Ours) | 75.06 | 83.30 | 84.65 | 89.74 | 92.68 |

Table S15 (continued): **Natural per-dataset performance** comparison of with various methods in few-shot setting in terms of classification accuracy (%).

| Dataset | Method | $K = 1$ | $K = 2$ | $K = 4$ | $K = 8$ | $K = 16$ |
|---------|--------|---------|---------|---------|---------|----------|
| StanfordCars | CLIP | | | 65.63 | | |
| | CLIP-Adapter | 67.10 | 68.97 | 71.13 | 76.47 | 80.90 |
| | Tip-Adapter | 66.47 | 68.03 | 70.08 | 71.61 | 74.08 |
| | Tip-Adapter-F | 67.88 | 71.39 | 74.59 | 78.34 | 83.09 |
| | Linear Probing | 33.70 | 48.93 | 62.34 | 71.21 | 80.10 |
| | LP++ | 66.31 | 70.41 | 74.57 | 76.12 | 80.77 |
| | CoOp | 67.10 | 69.37 | 72.73 | 76.57 | 78.89 |
| | CoCoOp | 67.92 | 68.77 | 69.10 | 70.19 | 71.68 |
| | KgCoOp | 67.03 | 68.13 | 71.98 | 73.53 | 74.87 |
| | ProGrad | 67.11 | 71.94 | 71.75 | 78.78 | 81.46 |
| | MaPLe | 66.60 | 71.60 | 75.30 | 79.47 | 83.57 |
| | CLIP-SVD (Ours) | 70.56 | 73.15 | 77.25 | 81.10 | 84.98 |
| Flowers102 | CLIP | | | 70.81 | | |
| | CLIP-Adapter | 54.83 | 54.83 | 54.83 | 56.08 | 56.50 |
| | Tip-Adapter | 80.74 | 86.59 | 89.62 | 92.15 | 93.68 |
| | Tip-Adapter-F | 86.63 | 90.21 | 91.50 | 94.98 | 96.18 |
| | Linear Probing | 72.50 | 85.48 | 91.65 | 95.89 | 97.36 |
| | LP++ | 86.22 | 90.38 | 92.96 | 95.25 | 96.44 |
| | CoOp | 71.98 | 76.12 | 82.56 | 84.17 | 87.64 |
| | CoCoOp | 82.20 | 88.47 | 91.14 | 94.37 | 96.10 |
| | KgCoOp | 74.63 | 79.47 | 90.69 | 89.53 | 92.90 |
| | ProGrad | 83.81 | 88.62 | 89.98 | 93.51 | 94.87 |
| | MaPLe | 83.30 | 88.93 | 92.67 | 95.80 | 97.00 |
| | CLIP-SVD (Ours) | 83.25 | 90.32 | 93.44 | 95.49 | 97.50 |
| FGVCAircraft | CLIP | | | 24.81 | | |
| | CLIP-Adapter | 27.60 | 29.60 | 31.10 | 37.20 | 42.67 |
| | Tip-Adapter | 27.22 | 28.08 | 30.19 | 34.12 | 37.64 |
| | Tip-Adapter-F | 29.58 | 32.51 | 35.15 | 40.61 | 45.59 |
| | Linear Probing | 20.13 | 23.78 | 29.46 | 37.18 | 42.85 |
| | LP++ | 29.11 | 32.32 | 34.31 | 38.83 | 41.46 |
| | CoOp | 15.03 | 26.53 | 26.73 | 33.18 | 40.93 |
| | CoCoOp | 13.20 | 30.87 | 31.29 | 29.57 | 37.63 |
| | KgCoOp | 26.90 | 28.07 | 32.47 | 34.97 | 36.27 |
| | ProGrad | 27.97 | 30.84 | 32.93 | 37.89 | 40.39 |
| | MaPLe | 26.73 | 30.90 | 34.87 | 42.00 | 48.40 |
| | CLIP-SVD (Ours) | 31.11 | 34.85 | 39.42 | 45.91 | 52.49 |
| SUN397 | CLIP | | | 62.60 | | |
| | CLIP-Adapter | 67.10 | 69.00 | 71.30 | 73.13 | 75.30 |
| | Tip-Adapter | 65.37 | 66.46 | 68.39 | 70.12 | 71.47 |
| | Tip-Adapter-F | 64.49 | 66.74 | 70.29 | 73.60 | 74.99 |
| | Linear Probing | 39.83 | 52.70 | 63.07 | 69.54 | 73.29 |
| | LP++ | 67.56 | 70.09 | 73.22 | 75.14 | 76.09 |
| | CoOp | 64.70 | 66.40 | 70.13 | 72.50 | 74.50 |
| | CoCoOp | 68.19 | 69.11 | 70.50 | 70.62 | 72.05 |
| | KgCoOp | 68.43 | 69.53 | 71.79 | 72.50 | 73.40 |
| | ProGrad | 64.54 | 68.51 | 71.17 | 72.91 | 75.00 |
| | MaPLe | 64.77 | 67.10 | 70.67 | 73.23 | 75.53 |
| | CLIP-SVD (Ours) | 70.37 | 71.93 | 73.75 | 75.16 | 76.85 |

Table S15 (continued): **Natural per-dataset performance** comparison of with various methods in few-shot setting in terms of classification accuracy (%).

| Dataset | Method | $K = 1$ | $K = 2$ | $K = 4$ | $K = 8$ | $K = 16$ |
|---|---|---|---|---|---|---|
| OxfordPets | CLIP | | | 89.10 | | |
| | CLIP-Adapter | 89.03 | 89.73 | 90.87 | 91.77 | 92.03 |
| | Tip-Adapter | 89.32 | 88.52 | 89.36 | 90.69 | 91.32 |
| | Tip-Adapter-F | 90.72 | 91.10 | 91.58 | 92.09 | 92.70 |
| | Linear Probing | 40.70 | 55.03 | 69.96 | 79.99 | 85.80 |
| | LP++ | 89.52 | 89.94 | 91.09 | 91.91 | 92.80 |
| | CoOp | 92.20 | 89.20 | 91.30 | 90.83 | 91.80 |
| | CoCoOp | 91.17 | 92.14 | 93.01 | 93.44 | 93.25 |
| | KgCoOp | 91.97 | 92.13 | 93.20 | 93.10 | 93.23 |
| | ProGrad | 89.01 | 90.55 | 93.21 | 92.18 | 92.13 |
| | MaPLe | 89.10 | 90.87 | 91.90 | 92.57 | 92.83 |
| | CLIP-SVD (Ours) | 92.67 | 92.09 | 92.83 | 92.57 | 93.77 |
| UCF101 | CLIP | | | 67.64 | | |
| | CLIP-Adapter | 69.67 | 74.10 | 77.30 | 81.87 | 84.53 |
| | Tip-Adapter | 68.92 | 71.50 | 74.07 | 75.65 | 77.35 |
| | Tip-Adapter-F | 73.27 | 76.11 | 80.24 | 82.24 | 84.50 |
| | Linear Probing | 48.24 | 61.08 | 70.87 | 77.51 | 81.27 |
| | LP++ | 72.36 | 75.13 | 79.49 | 82.17 | 83.75 |
| | CoOp | 24.96 | 25.89 | 23.85 | 26.23 | 28.48 |
| | CoCoOp | 25.42 | 28.85 | 30.66 | 21.78 | 24.86 |
| | KgCoOp | 72.93 | 74.83 | 78.40 | 80.03 | 81.43 |
| | ProGrad | 71.91 | 74.39 | 77.82 | 88.64 | 81.59 |
| | MaPLe | 71.83 | 74.60 | 78.47 | 81.37 | 85.03 |
| | CLIP-SVD (Ours) | 76.34 | 79.69 | 81.80 | 83.95 | 86.58 |
| Food101 | CLIP | | | 85.87 | | |
| | CLIP-Adapter | 85.90 | 86.10 | 86.47 | 86.67 | 86.83 |
| | Tip-Adapter | 85.14 | 86.03 | 85.99 | 86.42 | 86.36 |
| | Tip-Adapter-F | 86.01 | 86.26 | 86.48 | 86.74 | 87.24 |
| | Linear Probing | 44.63 | 62.66 | 72.96 | 78.98 | 83.69 |
| | LP++ | 83.23 | 86.12 | 85.94 | 86.76 | 87.28 |
| | CoOp | 82.07 | 80.80 | 82.58 | 83.37 | 85.17 |
| | CoCoOp | 86.10 | 86.21 | 86.64 | 86.92 | 87.19 |
| | KgCoOp | 86.27 | 86.60 | 86.59 | 86.90 | 87.03 |
| | ProGrad | 82.75 | 84.81 | 85.77 | 85.91 | 87.01 |
| | MaPLe | 80.50 | 81.47 | 81.77 | 83.60 | 85.33 |
| | CLIP-SVD (Ours) | 85.82 | 85.58 | 85.47 | 85.59 | 86.00 |
| **Average** | CLIP | | | 65.36 | | |
| | CLIP-Adapter | 67.87 | 70.20 | 72.65 | 76.92 | 79.86 |
| | Tip-Adapter | 68.89 | 70.42 | 72.69 | 74.41 | 76.44 |
| | Tip-Adapter-F | 70.62 | 73.08 | 75.75 | 78.51 | 81.15 |
| | Linear Probing | 45.77 | 56.92 | 66.79 | 73.43 | 78.39 |
| | LP++ | 70.35 | 72.93 | 75.77 | 77.94 | 80.32 |
| | CoOp | 68.09 | 70.13 | 73.59 | 76.45 | 79.01 |
| | CoCoOp | 66.95 | 67.63 | 71.98 | 72.92 | 75.02 |
| | KgCoOp | 69.51 | 71.57 | 74.48 | 75.82 | 77.26 |
| | ProGrad | 68.20 | 71.78 | 74.21 | 77.93 | 79.20 |
| | MaPLe | 69.27 | 72.58 | 75.37 | 78.89 | 81.79 |
| | CLIP-SVD (Ours) | 73.20 | 76.06 | 78.18 | 80.55 | 82.97 |

Table S16: **Biomedical per-dataset performance** comparison of CLIP-SVD with various methods in few-shot setting in terms of classification accuracy (%).

| Dataset | Method | $K = 1$ | $K = 2$ | $K = 4$ | $K = 8$ | $K = 16$ |
|---|---|---|---|---|---|---|
| BTMRI | BiomedCLIP | | | 56.79 | | |
| | CLIP-Adapter | 56.80 | 57.13 | 56.80 | 57.15 | 60.16 |
| | Tip-Adapter | 66.66 | 67.77 | 76.37 | 73.75 | 78.97 |
| | Tip-Adapter-F | 59.60 | 61.94 | 77.90 | 79.18 | 82.27 |
| | Standard LP | 62.24 | 72.45 | 75.98 | 77.63 | 81.24 |
| | LP++ | 64.72 | 71.69 | 75.48 | 77.11 | 81.61 |
| | CoOp | 63.82 | 68.82 | 74.68 | 79.27 | 82.37 |
| | CoCoOp | 59.47 | 64.14 | 67.83 | 71.69 | 78.45 |
| | KgCoOp | 63.33 | 70.16 | 75.40 | 79.79 | 81.07 |
| | ProGrad | 66.92 | 71.46 | 76.24 | 78.82 | 82.84 |
| | MaPLe | 38.01 | 37.02 | 42.36 | 50.75 | 56.22 |
| | BiomedCoOp | 65.08 | 70.57 | 77.23 | 78.55 | 83.30 |
| | CLIP-SVD (Ours) | 63.25 | 73.62 | 78.43 | 83.21 | 84.64 |
| BUSI | BiomedCLIP | | | 59.75 | | |
| | CLIP-Adapter | 61.44 | 61.01 | 61.72 | 61.86 | 63.55 |
| | Tip-Adapter | 62.71 | 61.44 | 59.03 | 55.93 | 68.78 |
| | Tip-Adapter-F | 61.86 | 56.35 | 64.54 | 68.50 | 71.89 |
| | Standard LP | 51.41 | 47.88 | 53.38 | 65.53 | 68.78 |
| | LP++ | 51.12 | 55.50 | 60.31 | 66.10 | 70.05 |
| | CoOp | 48.73 | 53.53 | 60.17 | 64.69 | 69.49 |
| | CoCoOp | 52.26 | 49.15 | 59.75 | 65.82 | 70.2 |
| | KgCoOp | 53.39 | 55.51 | 62.01 | 67.37 | 70.62 |
| | ProGrad | 46.33 | 49.15 | 62.29 | 64.83 | 71.47 |
| | MaPLe | 41.38 | 33.47 | 47.74 | 42.65 | 45.62 |
| | BiomedCoOp | 50.71 | 50.71 | 59.32 | 63.27 | 70.34 |
| | CLIP-SVD (Ours) | 61.58 | 63.56 | 68.12 | 73.87 | 74.29 |
| COVID-QU-Ex | BiomedCLIP | | | 43.8 | | |
| | CLIP-Adapter | 50.42 | 43.04 | 46.28 | 48.68 | 49.55 |
| | Tip-Adapter | 62.13 | 58.72 | 63.84 | 66.77 | 73.05 |
| | Tip-Adapter-F | 54.89 | 54.01 | 69.97 | 69.89 | 76.07 |
| | Standard LP | 49.91 | 48.06 | 60.55 | 68.29 | 71.98 |
| | LP++ | 46.41 | 56.42 | 62.32 | 66.19 | 72.79 |
| | CoOp | 58.82 | 58.37 | 67.03 | 74.66 | 76.37 |
| | CoCoOp | 69.36 | 68.80 | 63.70 | 69.36 | 74.52 |
| | KgCoOp | 61.68 | 54.68 | 65.91 | 74.86 | 75.65 |
| | ProGrad | 60.42 | 64.22 | 68.56 | 74.65 | 74.93 |
| | MaPLe | 35.86 | 38.99 | 33.32 | 36.43 | 40.89 |
| | BiomedCoOp | 72.64 | 71.53 | 73.28 | 76.26 | 78.72 |
| | CLIP-SVD (Ours) | 71.15 | 71.08 | 70.47 | 73.97 | 73.14 |
| CTKIDNEY | BiomedCLIP | | | 42.43 | | |
| | CLIP-Adapter | 47.17 | 41.94 | 42.19 | 44.64 | 47.28 |
| | Tip-Adapter | 45.85 | 51.65 | 55.33 | 69.89 | 73.38 |
| | Tip-Adapter-F | 46.68 | 58.99 | 60.18 | 75.24 | 82.07 |
| | Standard LP | 43.82 | 59.35 | 69.54 | 78.89 | 82.50 |
| | LP++ | 57.70 | 61.57 | 65.73 | 77.06 | 79.07 |
| | CoOp | 54.51 | 60.57 | 68.12 | 77.40 | 83.52 |
| | CoCoOp | 47.88 | 52.71 | 61.07 | 73.93 | 77.70 |
| | KgCoOp | 58.92 | 62.81 | 68.68 | 77.43 | 77.67 |
| | ProGrad | 54.65 | 64.66 | 67.90 | 78.23 | 81.13 |
| | MaPLe | 30.62 | 38.98 | 38.00 | 39.67 | 51.06 |
| | BiomedCoOp | 56.13 | 64.21 | 66.50 | 77.16 | 83.20 |
| | CLIP-SVD (Ours) | 55.64 | 58.88 | 74.31 | 78.33 | 86.34 |

Table S16 (continued): **Biomedical per-dataset performance** comparison of CLIP-SVD with various methods in few-shot setting in terms of classification accuracy (%).

| Dataset | Method | $K = 1$ | $K = 2$ | $K = 4$ | $K = 8$ | $K = 16$ |
|---|---|---|---|---|---|---|
| Kvasir | BiomedCLIP | | | 54.58 | | |
| | CLIP-Adapter | 54.83 | 54.83 | 54.83 | 56.08 | 56.50 |
| | Tip-Adapter | 56.72 | 60.94 | 69.61 | 69.13 | 74.22 |
| | Tip-Adapter-F | 59.19 | 64.22 | 69.94 | 75.86 | 78.00 |
| | Standard LP | 54.30 | 62.00 | 72.38 | 78.88 | 79.00 |
| | LP++ | 58.27 | 60.47 | 69.36 | 72.52 | 75.41 |
| | CoOp | 58.2 | 64.86 | 70.78 | 77.14 | 77.88 |
| | CoCoOp | 59.45 | 65.50 | 68.94 | 72.92 | 75.22 |
| | KgCoOp | 61.67 | 65.67 | 68.28 | 72.05 | 72.95 |
| | ProGrad | 60.78 | 64.70 | 70.00 | 76.03 | 75.88 |
| | MaPLe | 41.06 | 45.17 | 53.22 | 56.03 | 63.50 |
| | BiomedCoOp | 62.17 | 67.25 | 74.08 | 77.72 | 78.89 |
| | CLIP-SVD (Ours) | 65.08 | 72.08 | 74.92 | 80.08 | 82.83 |
| CHMNIST | BiomedCLIP | | | 30.65 | | |
| | CLIP-Adapter | 31.27 | 31.67 | 33.26 | 36.48 | 42.06 |
| | Tip-Adapter | 46.14 | 63.32 | 70.05 | 69.57 | 77.68 |
| | Tip-Adapter-F | 52.81 | 58.90 | 71.74 | 74.51 | 80.43 |
| | Standard LP | 58.44 | 64.42 | 71.07 | 76.30 | 80.34 |
| | LP++ | 57.18 | 60.61 | 67.79 | 72.40 | 78.32 |
| | CoOp | 57.34 | 59.68 | 68.66 | 75.00 | 79.63 |
| | CoCoOp | 49.07 | 50.82 | 58.58 | 66.58 | 72.16 |
| | KgCoOp | 59.02 | 60.06 | 68.77 | 69.50 | 73.58 |
| | ProGrad | 60.15 | 59.60 | 69.13 | 70.99 | 75.11 |
| | MaPLe | 48.05 | 57.76 | 65.87 | 68.88 | 71.99 |
| | BiomedCoOp | 59.82 | 59.79 | 71.19 | 74.78 | 79.05 |
| | CLIP-SVD (Ours) | 54.94 | 66.13 | 75.69 | 81.05 | 84.75 |
| LC25000 | BiomedCLIP | | | 50.03 | | |
| | CLIP-Adapter | 54.83 | 53.47 | 52.91 | 56.33 | 57.56 |
| | Tip-Adapter | 75.37 | 72.73 | 83.32 | 87.25 | 89.17 |
| | Tip-Adapter-F | 74.21 | 71.82 | 79.57 | 90.41 | 92.35 |
| | Standard LP | 74.50 | 78.40 | 85.30 | 90.24 | 92.77 |
| | LP++ | 63.05 | 71.42 | 82.61 | 89.14 | 92.58 |
| | CoOp | 71.90 | 76.55 | 84.66 | 87.50 | 92.19 |
| | CoCoOp | 63.66 | 71.76 | 77.44 | 85.57 | 87.38 |
| | KgCoOp | 71.80 | 75.18 | 82.10 | 84.63 | 86.79 |
| | ProGrad | 72.48 | 74.76 | 84.72 | 87.86 | 90.70 |
| | MaPLe | 67.13 | 69.80 | 76.73 | 82.19 | 86.73 |
| | BiomedCoOp | 77.56 | 77.74 | 85.60 | 88.77 | 92.68 |
| | CLIP-SVD (Ours) | 72.34 | 82.31 | 89.01 | 89.98 | 91.52 |
| RETINA | BiomedCLIP | | | 26.26 | | |
| | CLIP-Adapter | 25.49 | 25.49 | 26.07 | 25.84 | 26.05 |
| | Tip-Adapter | 26.52 | 31.07 | 43.42 | 48.08 | 54.23 |
| | Tip-Adapter-F | 39.53 | 33.07 | 47.37 | 56.07 | 62.85 |
| | Standard LP | 39.35 | 46.03 | 51.31 | 53.94 | 62.27 |
| | LP++ | 35.77 | 39.37 | 46.95 | 53.44 | 60.62 |
| | CoOp | 35.02 | 35.26 | 42.22 | 51.87 | 59.38 |
| | CoCoOp | 32.94 | 36.43 | 39.75 | 48.45 | 53.91 |
| | KgCoOp | 33.54 | 35.17 | 42.61 | 49.97 | 51.18 |
| | ProGrad | 33.49 | 36.49 | 43.09 | 52.26 | 50.47 |
| | MaPLe | 27.73 | 31.07 | 30.89 | 41.80 | 53.78 |
| | BiomedCoOp | 36.64 | 38.67 | 45.58 | 56.47 | 61.28 |
| | CLIP-SVD (Ours) | 41.46 | 44.93 | 52.63 | 62.33 | 68.06 |

Table S16 (continued): **Biomedical per-dataset performance** comparison of CLIP-SVD with various methods in few-shot setting in terms of classification accuracy (%).

| Dataset | Method | $K = 1$ | $K = 2$ | $K = 4$ | $K = 8$ | $K = 16$ |
|---------|--------|---------|---------|---------|---------|----------|
| KneeXray | BiomedCLIP | | | 29.53 | | |
| | CLIP-Adapter | 29.00 | 28.66 | 28.96 | 28.80 | 29.08 |
| | Tip-Adapter | 29.04 | 33.55 | 24.19 | 25.76 | 33.17 |
| | Tip-Adapter-F | 30.01 | 28.38 | 26.59 | 26.46 | 27.67 |
| | Standard LP | 26.02 | 26.57 | 27.83 | 22.20 | 23.97 |
| | LP++ | 21.25 | 26.40 | 28.92 | 23.75 | 26.38 |
| | CoOp | 24.96 | 25.89 | 23.85 | 26.23 | 28.48 |
| | CoCoOp | 25.42 | 28.85 | 30.66 | 21.78 | 24.86 |
| | KgCoOp | 29.07 | 28.14 | 22.44 | 23.37 | 24.80 |
| | ProGrad | 30.09 | 23.83 | 23.95 | 24.78 | 26.27 |
| | MaPLe | 23.41 | 22.67 | 22.56 | 24.78 | 25.87 |
| | BiomedCoOp | 36.13 | 37.72 | 35.91 | 37.7 | 39.69 |
| | CLIP-SVD (Ours) | 27.38 | 27.05 | 27.67 | 30.70 | 38.37 |
| OCTMNIST | BiomedCLIP | | | 30.00 | | |
| | CLIP-Adapter | 44.00 | 49.73 | 49.96 | 49.50 | 52.73 |
| | Tip-Adapter | 32.36 | 33.8 | 38.10 | 53.93 | 53.33 |
| | Tip-Adapter-F | 46.66 | 53.93 | 55.20 | 65.00 | 72.50 |
| | Standard LP | 47.25 | 54.21 | 61.00 | 65.85 | 69.40 |
| | LP++ | 47.24 | 53.18 | 59.02 | 63.69 | 68.35 |
| | CoOp | 52.63 | 53.57 | 53.37 | 63.67 | 65.47 |
| | CoCoOp | 49.33 | 50.93 | 48.57 | 55.40 | 60.67 |
| | KgCoOp | 50.63 | 50.53 | 52.97 | 61.03 | 62.80 |
| | ProGrad | 51.40 | 55.33 | 55.07 | 62.17 | 63.33 |
| | MaPLe | 26.63 | 34.00 | 30.17 | 30.53 | 33.63 |
| | BiomedCoOp | 51.83 | 55.03 | 54.73 | 58.87 | 66.93 |
| | CLIP-SVD (Ours) | 50.63 | 66.67 | 68.97 | 79.07 | 80.67 |
| **Average** | BiomedCLIP | | | 42.38 | | |
| | CLIP-Adapter | 45.53 | 44.70 | 45.30 | 46.54 | 48.46 |
| | Tip-Adapter | 50.35 | 53.50 | 58.33 | 62.01 | 67.60 |
| | Tip-Adapter-F | 52.55 | 54.17 | 62.30 | 68.12 | 68.12 |
| | Standard LP | 48.91 | 55.82 | 62.12 | 67.33 | 70.81 |
| | LP++ | 49.27 | 55.88 | 61.30 | 65.48 | 70.09 |
| | CoOp | 52.59 | 55.71 | 61.35 | 67.74 | 71.48 |
| | CoCoOp | 50.88 | 53.91 | 57.63 | 63.15 | 67.51 |
| | KgCoOp | 54.31 | 55.79 | 60.92 | 66.00 | 67.71 |
| | ProGrad | 53.67 | 56.42 | 62.10 | 67.06 | 69.21 |
| | MaPLe | 37.99 | 40.89 | 44.09 | 47.37 | 52.93 |
| | BiomedCoOp | 56.87 | 59.32 | 64.34 | 68.96 | 73.41 |
| | CLIP-SVD (Ours) | 56.35 | 62.63 | 68.02 | 73.26 | 76.46 |

Table S17: **Natural Base-to-novel generalization** comparison of CLIP-SVD with previous methods

| Dataset | | CLIP | CoOp | CoCoOp | KgCoOp | ProGrad | MaPLe | CLIP-SVD |
|---|---|---|---|---|---|---|---|---|
| Average on 11 datasets | Base | 69.34 | 82.69 | 80.47 | 80.73 | 82.48 | 82.28 | **84.38** |
| | Novel | 74.22 | 63.22 | 71.69 | 73.60 | 70.75 | 75.14 | **76.29** |
| | HM | 71.70 | 71.66 | 75.83 | 77.00 | 76.16 | 78.55 | **80.13** |
| ImageNet | Base | 72.43 | 76.47 | 75.98 | 75.83 | 77.02 | 76.66 | **78.01** |
| | Novel | 68.14 | 67.88 | 70.43 | 69.96 | 66.66 | 70.54 | **70.71** |
| | HM | 70.22 | 71.92 | 73.10 | 72.78 | 71.46 | 73.47 | **74.18** |
| Caltech101 | Base | 96.84 | 98.00 | 97.96 | 97.72 | 98.02 | 97.74 | **98.54** |
| | Novel | 94.00 | 89.81 | 93.81 | **94.39** | 93.89 | 94.36 | 94.00 |
| | HM | 95.40 | 93.73 | 95.84 | 96.03 | 95.91 | 96.02 | **96.21** |
| OxfordPets | Base | 91.17 | 93.67 | 95.20 | 94.65 | 95.07 | 95.43 | **96.28** |
| | Novel | 97.26 | 95.29 | 97.69 | **97.77** | 97.63 | 97.76 | 97.60 |
| | HM | 94.12 | 94.47 | 96.43 | 96.18 | 96.33 | 96.58 | **96.93** |
| Stanford Cars | Base | 63.37 | 78.12 | 70.49 | 71.76 | 77.68 | 72.94 | **78.89** |
| | Novel | 74.89 | 60.40 | 73.59 | **75.04** | 68.63 | 74.00 | 74.03 |
| | HM | 68.65 | 68.13 | 72.01 | 73.36 | 72.88 | 73.47 | **76.38** |
| Flowers102 | Base | 72.08 | **97.60** | 94.87 | 95.00 | 95.54 | 95.92 | 96.30 |
| | Novel | **77.80** | 59.67 | 71.75 | 74.73 | 71.87 | 72.46 | 76.19 |
| | HM | 74.83 | 74.06 | 81.71 | 83.65 | 82.03 | 82.56 | **85.07** |
| Food101 | Base | 90.10 | 88.33 | 90.70 | 90.50 | 90.37 | **90.71** | 90.52 |
| | Novel | 91.22 | 82.26 | 91.29 | 91.70 | 89.59 | **92.05** | 91.74 |
| | HM | 90.66 | 85.19 | 90.99 | 91.09 | 89.98 | **91.38** | 91.13 |
| FGVC Aircraft | Base | 27.19 | 40.44 | 33.41 | 36.21 | 40.54 | 37.44 | **44.00** |
| | Novel | 36.29 | 22.30 | 23.71 | 33.55 | 27.57 | 35.61 | **37.01** |
| | HM | 31.09 | 28.75 | 27.74 | 34.83 | 32.82 | 36.50 | **40.20** |
| SUN397 | Base | 69.36 | 80.60 | 79.74 | 80.29 | 81.26 | 80.82 | **82.56** |
| | Novel | 75.35 | 65.89 | 76.86 | 76.53 | 74.17 | 78.70 | **78.98** |
| | HM | 72.23 | 72.51 | 78.27 | 78.36 | 77.55 | 79.75 | **80.73** |
| DTD | Base | 53.24 | 79.44 | 77.01 | 77.55 | 77.35 | 80.36 | **82.91** |
| | Novel | 59.90 | 41.18 | 56.00 | 54.99 | 52.35 | 59.18 | **65.42** |
| | HM | 56.37 | 54.24 | 64.85 | 64.35 | 62.45 | 68.16 | **73.13** |
| EuroSAT | Base | 56.48 | 92.19 | 87.49 | 85.64 | 90.11 | 94.07 | **94.53** |
| | Novel | 64.05 | 54.74 | 60.04 | 64.34 | 60.89 | 73.23 | **74.03** |
| | HM | 60.03 | 68.69 | 71.21 | 73.48 | 72.67 | 82.35 | **83.03** |
| UCF101 | Base | 70.53 | 84.69 | 82.33 | 82.89 | 84.33 | 83.00 | **85.66** |
| | Novel | 77.50 | 56.05 | 73.45 | 76.67 | 74.94 | 78.66 | **79.45** |
| | HM | 73.85 | 67.46 | 77.64 | 79.65 | 79.35 | 80.77 | **82.44** |

Table S18: **Biomedical Base-to-novel generalization** comparison of CLIP-SVD with previous methods

| Dataset | | CLIP | CoOp | CoCoOp | KgCoOp | ProGrad | MaPLe | BiomedCoOp | CLIP-SVD |
|---|---|---|---|---|---|---|---|---|---|
| Average on 9 datasets | Base | 49.27 | 76.71 | 75.52 | 71.90 | 75.69 | 65.40 | 78.60 | **82.64** |
| | Novel | 67.17 | 65.34 | 67.74 | 65.94 | 67.33 | 49.51 | 73.90 | **74.31** |
| | HM | 55.23 | 68.80 | 69.11 | 67.22 | 69.86 | 53.10 | 74.04 | **78.25** |
| BTMRI | Base | 40.88 | 82.25 | 77.88 | 78.00 | 82.13 | 66.17 | 82.42 | **88.83** |
| | Novel | 96.18 | 94.51 | 94.84 | 95.05 | 94.98 | 49.58 | **96.84** | 94.98 |
| | HM | 57.37 | 87.95 | 85.53 | 85.69 | 88.09 | 56.69 | 89.05 | **91.80** |
| COVID-QU-Ex | Base | 53.96 | 75.92 | **77.28** | 75.42 | 75.19 | 61.36 | 75.91 | 75.39 |
| | Novel | 89.43 | 90.07 | 87.61 | 89.61 | 90.34 | 71.25 | **91.63** | 89.56 |
| | HM | 67.31 | 82.39 | 82.12 | 81.90 | 82.07 | 65.94 | **83.03** | 81.87 |
| CTKIDNEY | Base | 38.55 | 82.24 | 81.96 | 81.67 | 83.86 | 63.99 | 86.93 | **88.31** |
| | Novel | 52.99 | 67.92 | 56.56 | **58.45** | 63.01 | 63.51 | **78.94** | 68.02 |
| | HM | 44.63 | 74.40 | 66.93 | 68.14 | 71.96 | 63.75 | **82.74** | 76.85 |
| Kvasir | Base | 75.00 | 86.22 | 85.94 | 81.56 | 82.89 | 76.66 | 86.50 | **86.78** |
| | Novel | **60.50** | 58.06 | 53.95 | 59.00 | 60.45 | 26.17 | 61.83 | **61.89** |
| | HM | 66.97 | 69.39 | 66.29 | 68.47 | 69.91 | 39.02 | 72.11 | **72.25** |
| CHMNIST | Base | 37.63 | 89.41 | 87.77 | 75.45 | 82.98 | 89.19 | 88.87 | **90.38** |
| | Novel | 40.69 | 35.11 | 42.51 | 38.70 | 44.19 | 23.76 | **42.73** | 35.42 |
| | HM | 39.10 | 50.42 | 57.28 | 51.16 | 57.67 | 37.52 | **57.71** | 50.89 |
| LC25000 | Base | 59.73 | 90.12 | 88.33 | 88.13 | 90.29 | 87.16 | 93.77 | **96.68** |
| | Novel | 87.60 | 87.55 | 95.02 | 86.44 | 85.47 | 52.66 | **97.00** | 96.36 |
| | HM | 71.03 | 88.82 | 91.55 | 87.28 | 87.81 | 65.65 | 95.36 | **96.52** |
| RETINA | Base | 45.18 | 70.98 | 66.88 | 60.77 | 68.77 | 57.40 | 68.46 | **84.62** |
| | Novel | 55.28 | 56.90 | 65.56 | 54.91 | 58.43 | 53.33 | 67.72 | **84.88** |
| | HM | 49.72 | 63.16 | 66.21 | 57.69 | 63.18 | 55.29 | 68.09 | **84.75** |
| KneeXray | Base | 35.89 | 38.28 | 34.08 | 37.94 | 40.88 | 36.44 | **44.23** | 43.73 |
| | Novel | 71.90 | 47.69 | 63.14 | 61.19 | 59.12 | 55.35 | **78.35** | 76.52 |
| | HM | 47.88 | 42.47 | 44.27 | 46.84 | 48.34 | 43.95 | **56.54** | 55.65 |
| OCTMNIST | Base | 56.60 | 75.00 | 79.60 | 68.20 | 74.20 | 50.27 | 80.33 | **89.00** |
| | Novel | 50.00 | 50.23 | 50.47 | 50.13 | 50.02 | 50.00 | 50.07 | **61.13** |
| | HM | 53.10 | 60.17 | 61.77 | 57.79 | 59.76 | 50.13 | 61.69 | **72.48** |

