# OpenReview forum: "CLIP-SVD: Efficient and Interpretable Vision–Language Adaptation via Singular Values"
_TMLR — Accepted by TMLR_

### Review · Reviewer_AaTG · 2025-10-31

**Summary Of Contributions:**

The article proposes CLIP-SVD, a parameter-efficient fine-tuning type (PEFT) method for CLIP-style vision-language models adaptation by performing singular value decomposition to trainable weights with the subsequent freezing of singular vectors of each weight matrix and updating only non-negative singular values. This method is applied to both image and text encoders represented by superpositions of Multi-Head Self-Attention (MHSA) layers and Multi-Layered Perceptron (MLP) layers. Each pretrained trainable matrix $W$ is decomposed using SVD into product $W = U\Sigma R^\top$, where $U$ and $R$ are orthogonal (unitary) matrices and $\Sigma$ is a rectangular diagonal matrix with non-negative elements on the diagonal. The framework freezes $U$ and $R$ while keeping diagonal weights of $\Sigma$ trainable which satisfies the notion of PEFT paradigm. The authors evaluate this framework on 11 natural and 10 biomedical classification datasets under few-shot regimes, report strong results on base-to-novel and cross-dataset transfer tasks, and perform an interpretability study using TextSpan method. The authors also proposed a GPT-4-generated biomedical caption corpora to analyze attention-based semantics after adaptation using the aforementioned TextSpan method.

The methodology of SVD incorporation into CLIP is clearly described, the introduced framework is extensively tested on natural and biomedical multimodal text-image domains. The experiments cover few-shot learning setting, varying the number of labled samples, $K\in\\{1, 2, 4, 8, 16\\}$. The other tested tasks are Base-to-Novel Generalization, Cross-dataset Evaluation. The framework is tested against existinmg SOTA demonstrating competitive results. Base-to-Novel Generalization task results demonstrate harmonic mean improvement over strong baselines, Cross-dataset Transfer task results are on par with existing SOTA with a slight improvement against MaPLe framework. The authors performed ablation studies of their framework by alternating types of trainable weights (query, keys, values, MHSA output, MLP), modalities used (text, images, both), the results of these ablations are consistent, without contradictions: the more singular layers trained &mdash; the better quality of the framework, both texts and images work together best. The application of TextSpan increases interpretability of the learned MHSA heads and layers.

**Additional Comments:**

Current decision: minor or moderate corrections are needed.

**Audience:**

Yes

**Audience Explanation:**

The paper is well-structured, the results are consisted of rather interesting SOTA values on practically important ML tasks yet simple by design.

**Broader Impact Concerns:**

No ethical concerns implied.

**Claims And Evidence:**

Yes

**Claims Explanation:**

The method description is consistent and technically sound. Claims of broad SOTA performance in few-shot, base-to-novel, and cross-dataset are borne out by tables, albeit with modest margins in some settings.

**Requested Changes:**

Despite featuring good qualities, the paper itself has flawed details, listed below:

* The way how trainable weights are trained and updated is not clear enough, e.g. how the non-negativity of singular values ispreserved. How SVD results are stored and recomputed during optimization and inference? Consider adding a table with exact trainable parameter counts per layer.
* Consider adding a small paragraph describing the models used in the experiments, how these models are pretrained (or randomly initialized)?
* Has the correctness of GPT-4 generated datasets been checked, how reliable the results of experiments on (partially) artificially generated data are? The paper would only benefit from at least sampled expert rating to validate descriptions/head rankings. Consider adding human-study validation or quantitative alignment metrics.
* The authors should strengthen the novelty of their results especially compared to SVF and LoRA.

The concerns listed above should be addressed carefully.

---

> ### Author Response · Authors · 2026-02-02
> **Response to Reviewer AaTG (1/3)**
>
> We thank the reviewer for the thorough and technically detailed review, as well as for the constructive suggestions aimed at improving clarity, rigor, and positioning of our work. We appreciate the careful assessment of both the methodology and experiments, and we address each of the raised concerns below. The reviewer’s feedback has helped us substantially strengthen the presentation and completeness of the paper.
>
> ---
>
> 1. **"The way how trainable weights are trained and updated is not clear enough, e.g. how the non-negativity of singular values is preserved."**
>
>    To clarify the update mechanism: for each pretrained weight matrix
>
>    $$
>    W \in \mathbb{R}^{d_{\text{out}} \times d_{\text{in}}}
>    $$
>
>    we compute a truncated singular value decomposition
>
>    $$
>    W \approx U_r \mathrm{diag}(\sigma) V_r^\top
>    $$
>
>    where
>    $U_r \in \mathbb{R}^{d_{\text{out}} \times r}$ and
>    $V_r \in \mathbb{R}^{d_{\text{in}} \times r}$
>    contain the top $r$ left and right singular vectors, and
>    $\sigma \in \mathbb{R}^r$ contains the corresponding singular values.
>
>    During fine-tuning, we **freeze** $U_r$ and $V_r$, and treat only the singular values as trainable parameters. The adapted weight therefore becomes
>
>    $$
>    W_{\theta} = U_r \mathrm{diag}(s) V_r^\top
>    $$
>
>    where $s \in \mathbb{R}^r$ is updated directly through backpropagation.
>
>    We do not explicitly enforce $s \ge 0$. Allowing negative entries is mathematically harmless: any sign change in a singular value can be absorbed by flipping the sign of the corresponding singular vector, yielding an equivalent outcome. Thus, the parameterization remains well-posed while constraining learning to the spectral subspace defined by the pretrained model. Notably, this design choice is aligned with the official implementation of SVF (Sun et al., 2022), which similarly does not impose non-negativity constraints during spectral adaptation. We have added a detailed explanation in **Section 3.2**.
>
> ---
>
> 2. **"How SVD results are stored and recomputed during optimization and inference?"**
>
>    During optimization and inference, the model no longer uses the original full weight matrix $W$. Instead, each linear layer is reparameterized once at initialization by its SVD factorization (see our response to **Reviewer Vrgs, Comment 2**): the fixed singular vectors $U_r$ and $V_r$, together with the trainable singular values $s$.
>
>    Every forward pass computes the effective weight on the fly as
>
>    $$
>    W_{\theta} = U_r \mathrm{diag}(s) V_r^\top
>    $$
>
>    and this factorized form is used consistently throughout training and deployment.
>
>    If a standard linear weight is ever needed (e.g., for exporting or merging the adapter back into the backbone), the reconstructed matrix
>
>    $$
>    \widehat{W} = U_r \mathrm{diag}(s) V_r^\top
>    $$
>
>    can be explicitly formed without affecting the learned parameters. Thus, the SVD components themselves constitute the stored and updated representation of each adapted layer. We have added a detailed explanation in **Section 3.2**.

---

> > ### Author Response · Authors · 2026-02-02
> > **Response to Reviewer AaTG (2/3)**
> >
> > 3. **"Consider adding a table with exact trainable parameter counts per layer."**
> >
> >     Thank you for this suggestion. We have added **Table S11** in **Appendix J (Computational Cost)**, which provides the exact trainable parameters for each layer for both CLIP and BiomedCLIP.
> >
> >
> >     **Table R3: Trainable parameters in CLIP-SVD (full-rank SVD)**
> >
> >     | Model / Encoder                      | Hidden Size $D$ | MHSA Params ($4D$) | MLP Params ($2D$) | **Total per Layer ($6D$)** | **Total per Encoder (12 Layers = $72D$)** |
> >     |--------------------------------------|------------------|---------------------|--------------------|-----------------------------|---------------------------------------------|
> >     | **CLIP Text Encoder**                | $512$            | $2,048$           | $1,024$          | $3,072$                   | $36,864$                                  |
> >     | **CLIP ViT-B/16 Vision Encoder**     | $768$            | $3,072$           | $1,\!536$          | $4,608$                   | $55,296$                                  |
> >     | **CLIP Total (Text + Vision)**       | —                | —                   | —                  | —                           | **$92,160$**                              |
> >     |                                      |                  |                     |                    |                             |                                             |
> >     | **BiomedCLIP Text (PubMedBERT)**     | $768$            | $3,072$           | $1,536$          | $4,608$                   | $55,296$                                  |
> >     | **BiomedCLIP Vision (ViT-B)**        | $768$            | $3,072$           | $1,536$          | $4,608$                   | $55,296$                                  |
> >     | **BiomedCLIP Total (Text + Vision)** | —                | —                   | —                  | —                           | **$110,592$**                             |
> >
> > ----
> >
> > 4. **"Consider adding a small paragraph describing the models used in the experiments, how these models are pretrained (or randomly initialized)?"**
> >
> >     For natural domain experiments, we use the standard ViT-B/16 CLIP model pretrained on 400M image–text pairs from the WebImageText (WIT) corpus, where the vision and text encoders are jointly trained to align images with natural-language descriptions. For biomedical-domain experiments, we adopt BiomedCLIP, a domain-specialized VLM pretrained on 15 million biomedical image–caption pairs sourced from PubMed Central (PMC) scientific articles, with the  ViT-B/16 image encoder and the PubMedBERT text encoder. All compared adaptation methods, including CoOp, CoCoOp, LoRA/CLIP-LoRA, MaPLe, and BiomedCoOp, are initialized directly from these same pretrained encoders, ensuring a consistent and fair comparison. For methods that use prompt tuning, we initialized the prompts with the embedding of "a photo of a", while other adaptation techniques were randomly initialized. Each method differs only in the adaptation strategy applied on top of these frozen pretrained weights, while CLIP-SVD fine-tunes the singular values of the weight matrices solely, preserving the pretrained geometry of both CLIP and BiomedCLIP. We have added a detailed explanation in **Section 4.1** under **Implementation Details**.

---

> > > ### Author Response · Authors · 2026-02-02
> > > **Response to Reviewer AaTG (3/3)**
> > >
> > > 5. **"Has the correctness of GPT-4 generated datasets been checked, how reliable the results of experiments on (partially) artificially generated data are? The paper would only benefit from at least sampled expert rating to validate descriptions/head rankings. Consider adding human-study validation or quantitative alignment metrics."**
> > >
> > >     We thank the reviewer for this insightful comment. To ensure the correctness of the GPT-4 generated text prompts, we conducted a human validation step performed by a researcher with extensive experience in biomedical image analysis. The validator routinely works with medical imaging datasets and associated clinical terminology. They reviewed *all generated prompts*, corrected two minor inconsistencies, and further organized the full set into **seven semantically coherent categories**, which we describe in detail in **Appendix B**, where we also add **Figure S1**. In addition to human validation, we incorporated an independent quantitative assessment using three separate large language models (Gemini 3, GPT-5, and Qwen3) acting as judges, which has been recently shown to be effective, rivaling human crowd accuracy (Schoenegger et al., 2024). Each model independently scored, on a 1–5 scale, the relevance between the role description of each attention head/layer, which we extracted using TextSpan (Gandelsman et al., 2023), and the domain-specific characteristics of the dataset. The average score across all heads and all judges was **3.97**, indicating strong alignment between the adapted attention patterns and the underlying domain features. A full explanation of the evaluation protocol and detailed results are now provided in **Appendix G** and **Table S9**. These added validations directly address concerns regarding reliability and quantitative metrics for the domain alignment quality of the generated biomedical corpus.
> > >
> > > ---
> > >
> > > 6. **"The authors should strengthen the novelty of their results especially compared to SVF and LoRA."**
> > >
> > >     Thank you for your comment. We highlight our novelty in **Section 5** of the manuscript. SVF (Sun et al., 2022) focuses on CNNs and single-modal classification, whereas CLIP-SVD is the **first SVD-based adaptation application for multi-modal Transformers**, enabling principled modulation of key Attention components in both the vision and text encoders. Unlike LoRA (Hu et al., 2021), which introduces new low-rank matrices that can distort the pretrained geometry (e.g., shifting singular vectors) as suggested by Shuttlewoth et al. (2025) , CLIP-SVD **preserves CLIP’s semantic structure** by freezing the singular vectors and tuning only the singular values, yielding more stable and data-efficient adaptation. A key distinction of our approach is its **subspace-level interpretability**, which neither SVF nor LoRA addresses: because the semantic directions (U, R) remain fixed, and because CLIP provides rich text embeddings, we can directly connect changes in singular values to **language-grounded semantic shifts**, revealing which pretrained linguistic–visual directions are amplified, suppressed, or repurposed during adaptation. This interpretability, combined with superior generalization in the biomedical few-shot setting, differentiates CLIP-SVD from prior PEFT methods. We have further revised **Section 5** to better highlight these contributions. We also revised the paper title to more accurately capture the intended application and scope of the manuscript.

---

### Review · Reviewer_1w3M · 2025-10-31

**Summary Of Contributions:**

This paper proposes CLIP-SVD, a method for efficiently adapting a pretrained vision-language models like CLIP to new tasks, using very little data. The idea is simple yet powerful: instead of adding new modules or performing complete fine-tuning, they apply SVD to the pre-trained weight matrices, freeze the singular vectors and only tune the singular values.
In essence, they are simply "recalibrating" the importance of the directions already learned by the model, without distorting them. The result is that, with only 0.04% of trainable parameters, it beats many more complex methods on 21 natural and biomedical datasets.
The interpretability part with TextSpan is very interesting and provides a sense of what really happens during adaptation. This is solid work, well-motivated, and with convincing experiments.

Strengths: The results are robust and convincing, especially its ability to generalize well to both natural and biomedical domains while training a minimal fraction of parameters.

A small tip: For a reader who isn't super familiar with the field, some concepts, such as CLIP's internal architecture or the details of transformers, are a bit taken for granted. The paper is already very clear, but a little more effort to explain these fundamentals in a future draft could make it accessible to an even wider audience.

**Audience:**

Yes

**Audience Explanation:**

This paper will be highly relevant to the TMLR audience, particularly researchers working on parameter-efficient fine-tuning, vision-language models, and model interpretability. Its novel SVD-based adaptation method and strong empirical results across both natural and biomedical domains offer valuable insights for anyone adapting large pre-trained models. The work also provides a practical and efficient solution for few-shot learning scenarios that will interest both academic researchers and practitioners.

**Claims And Evidence:**

Yes

**Claims Explanation:**

-The paper provides an extensive empirical evaluation across a massive and diverse benchmark of 21 datasets (11 natural, 10 biomedical). This scale alone makes the evidence highly convincing, as it demonstrates robustness across vastly different domains.

-The performance claims are directly supported by head-to-head comparisons with a wide array of state-of-the-art baselines across multiple few-shot settings. The consistent outperformance of these methods in key tables is compelling.

-The use of TextSpan to link mathematical changes (singular value shifts) to semantic shifts in attention heads is a sophisticated and highly convincing way to support the mechanism behind their method.

**Requested Changes:**

I have no critical changes to request for the acceptance of this submission

The authors might consider adding a brief statement in the conclusion committing to the public release of their code upon publication

---

> ### Author Response · Authors · 2026-02-02
> **Response to Reviewer 1w3M**
>
> We thank the reviewer for the thoughtful and encouraging evaluation of our work. We appreciate the positive feedback on the empirical results, parameter efficiency, and interpretability analysis, as well as the constructive suggestion regarding code release. Below, we address the requested update.
>
> ----
>
> 1. **"I have no critical changes to request for the acceptance of this submission. The authors might consider adding a brief statement in the conclusion committing to the public release of their code upon publication"**
>
>     Thank you for the positive assessment of our work. In response to the suggestion, we have updated the conclusion to explicitly state our commitment to publicly releasing the full codebase, including the new biomedical caption corpus, upon publication. We believe this addition further strengthens the transparency and reproducibility of our approach.

---

### Review · Reviewer_Vrgs · 2026-01-24

**Summary Of Contributions:**

The paper proposes "CLIP-SVD," a parameter-efficient fine-tuning  method for adapting Vision-Language Models (specifically CLIP) to downstream tasks. The core methodology involves performing SVD on the weight matrices of the pre-trained model, freezing the singular vectors, and only fine-tuning the singular values. This approach is applied to the Multi-Head Self-Attention and MLP blocks of both the image and text encoders. The authors evaluate this method on few-shot classification, base-to-novel generalization, and domain generalization across 11 natural image datasets and 10 biomedical datasets. They also utilize an existing interpretability tool (TextSpan) to analyze changes in attention heads.

**Additional Comments:**

Strengths:

Comprehensive Evaluation: The paper performs extensive experiments across a wide variety of datasets (both natural and biomedical) and settings (few-shot, domain shift), providing a solid empirical benchmark.
Parameter Efficiency: The method demonstrates competitive performance while updating a very small fraction of parameters (0.04%).
Interpretability Attempt: The inclusion of the TextSpan analysis provides some insight into how the adaptation alters the model's semantic focus.

Weaknesses:

Lack of Methodological Innovation: The core technique is a direct application of Singular Value Fine-tuning (SVF), which has already been proposed and explored in prior works for CNNs and LLMs. The application to CLIP is a straightforward extension without significant technical modification or theoretical insight specific to multi-modal learning.
Incremental Contribution: The paper frames the method as a "novel technique," but it is essentially an empirical study of an existing technique applied to a different architecture. There is no specific exploitation of the vision-language modality structure other than applying the method to both encoders.
Marginal Improvements: In several benchmarks (e.g., Table 4, Natural domain base-to-novel), the improvement over methods like CLIP-LoRA is relatively marginal (e.g., HM 80.13% vs 79.18%), raising questions about whether the complexity of SVD decomposition is justified over standard low-rank adaptations.

**Audience:**

Yes

**Audience Explanation:**

Despite the lack of novelty, TMLR's audience includes practitioners interested in efficient model adaptation. Knowing that SVF (a technique known in other domains) works effectively for CLIP and particularly well in biomedical few-shot settings is a useful empirical data point. The extensive benchmarking on biomedical datasets is of value to those working in medical imaging.

**Claims And Evidence:**

Yes

**Claims Explanation:**

The authors provide extensive tabular results comparing their method against relevant baselines (CoOp, MaPLe, CLIP-Adapter, CLIP-LoRA). The experimental setup follows standard protocols (16-shot, etc.) established in prior literature (Zhou et al., 2022b). The results consistently show that CLIP-SVD performs comparably to or slightly better than state-of-the-art PEFT methods. The ablation studies (Table 7) support the decision to tune specific components. While the novelty is limited, the empirical evidence for the method's effectiveness in this specific setting is presented clearly.

**Requested Changes:**

Reframe Claims of Novelty:
The paper currently claims to present a "novel multi-modal and parameter-efficient adaptation technique." This is misleading. The technique (SVF) is not novel; the application to CLIP is the contribution. The authors must rewrite the Abstract and Introduction to explicitly acknowledge that SVF is an existing technique (citing Sun et al. and Meng et al. prominently in the intro) and clarify that this work is an empirical adaptation and evaluation of SVF in the VLM context. The contribution should be framed as "The first extensive empirical evaluation of SVF for Vision-Language Models," rather than a new method.

Computational Cost Clarification:
While the method is parameter-efficient regarding trainable parameters, SVD decomposition of large weight matrices is computationally expensive and memory-intensive compared to simple random initialization used in LoRA or adapters. The authors should add a discussion or analysis regarding the "pre-processing" cost of performing SVD on all weights before training begins. Does this overhead negate the efficiency gains in low-resource settings?

Baselines and Fairness: In the comparisons (e.g., Table 2), the authors compare 0.04% trainable parameters (CLIP-SVD) against methods with significantly higher counts. While this highlights efficiency, it would be strengthening to see a comparison where baselines (like LoRA) are constrained to a similar parameter budget (by reducing rank) to see if SVF is truly superior per-parameter, or just effective because full-rank singular value tuning covers enough capacity.

---

> ### Author Response · Authors · 2026-02-02
> **Response to Reviewer Vrgs (1/2)**
>
> We thank the reviewer for their careful reading of our manuscript and for the constructive and balanced feedback. We appreciate the reviewer’s recognition of the empirical value of our work, particularly in the few-shot and biomedical settings, as well as the thoughtful suggestions regarding novelty framing, computational cost, and fair comparisons. We address each requested change in detail below and have revised the paper accordingly, which has helped us significantly improve both clarity and presentation.
>
> ----
>
> 1. **"The paper currently claims to present a novel multi-modal and parameter-efficient adaptation technique, which is misleading since SVF is not novel."**
>
>     We thank the reviewer for pointing this out. We have revised the **Abstract** and **Introduction** to clearly acknowledge that Singular Value Fine-tuning (SVF) is an existing technique proposed by Sun et al. and further explored by Meng et al., and and we now frame our contribution as an **empirical adaptation and systematic evaluation of SVF in the vision–language model (VLM) setting**. While prior SVF work is limited to **CNN-based, single-modal classification**, extending spectral adaptation to **CLIP-style multi-modal Transformers** is non-trivial. CLIP-SVD applies SVF jointly to the vision and text encoders, enabling principled modulation of Transformer components while preserving pretrained geometry by freezing singular vectors and tuning only singular values, unlike LoRA-style adapters that introduce new trainable subspaces. We therefore position this work as the **first comprehensive empirical study of SVF for multi-modal Transformers**, covering natural and biomedical domains, few-shot learning, and generalization. This formulation also enables language-grounded **subspace-level interpretability**, linking changes in singular values to task-related semantic shifts, which is explained in detail in **Section 4.6** and **Section 5 (Discussion)**. All novelty claims have been updated accordingly, and prior SVF work is now prominently discussed in the paper. We also revised the paper title to more accurately capture the intended application and scope of the manuscript.
>
> ----
>
> 2. **"The computational and memory cost of performing SVD on large weight matrices should be clarified."**
>
>      We thank the reviewer for raising this point regarding the computational cost of SVD. We have added a discussion in **Appendix J (Computational Cost)** with **Table S12** in the revised paper quantifying the runtime and GPU memory usage of the SVD decomposition for CLIP and BiomedCLIP and pointed to this in **Section 3.2** of the manuscript. Importantly, this cost is incurred **only once** as a **model-level preprocessing step** and is **independent of the downstream task or adaptation setting**. The pretrained backbones are decomposed a single time (see our response to **Comment 2** by **Reviewer AaTG** for more details), after which the resulting SVD-parameterized zero-shot models can be **saved and reused across all experiments**, with no further SVD computation required. As shown in **Table R1**, this one-time cost is modest (**12.16s for CLIP**, **15.23s for BiomedCLIP**). Once preprocessing is complete, training and inference incur **no additional overhead**, since only the singular values are optimized. Although this preprocessing is more expensive than random initialization in LoRA, it is shared across tasks and negligible in practical multi-task adaptation scenarios, preserving the efficiency of CLIP-SVD.
>
>
>      **Table R1: Pre-processing SVD Computational Cost and Memory Usage**
>       Method        | Time per Block (ms) | Total Time (s) | Peak GPU Memory (GB) |
>     |---------------|---------------------------|--------------------|----------------------|
>     | CLIP          | 84.46                    | 12.16              | 1.83              |
>     | BiomedCLIP    | 103.58                     | 15.23              | 2.46               |

---

> ### Author Response · Authors · 2026-02-02
> **Response to Reviewer Vrgs (2/2)**
>
> 3. **“Baselines should be compared under a similar parameter budget to ensure fairness.”**
>
>     Thank you for this suggestion. To explicitly address parameter fairness, we have added **CLIP-LoRA with rank = 1** to **Table 2** in the main paper, which uses a **comparable number of trainable parameters as CLIP-SVD** (see our response to **Comment 3** by **Reviewer AaTG** regarding the trainable-parameter formulation and counts). For reference, CLIP-LoRA with rank = 2 uses **approximately twice the number of trainable parameters**, as detailed in **Tables S7 and S8** of the supplementary material. As shown in **Table R2**, reducing CLIP-LoRA to rank = 1 leads to a slight drop in few-shot performance compared to rank = 2, and **CLIP-SVD still consistently outperforms CLIP-LoRA even when both methods operate under an identical parameter budget**, demonstrating superior per-parameter effectiveness. This result highlights the advantage of spectral adaptation, which preserves the pretrained subspace while enabling efficient and stable fine-tuning.
>
>     **Table R2: Natural-domain few-shot classification accuracy (%)**
>
>     | Method                  | K = 1 | K = 2 | K = 4 | K = 8 | K = 16 |
>     |-------------------------|-------|-------|-------|-------|--------|
>     | CLIP-LoRA (rank = 1)    | 72.16 | 75.29 | 77.28 | 80.02 | 82.83 |
>     | CLIP-LoRA (rank = 2)    | *72.20* | *75.41* | *77.32* | *80.10* | *82.89* |
>     | **CLIP-SVD (Ours)**     | **73.20** | **76.06** | **78.18** | **80.55** | **82.97** |
>
> ----
>
> 4. **"The contribution appears incremental, with marginal improvements over LoRA in some benchmarks."**
>
>    We thank the reviewer for this thoughtful comment. We have revised **Section 5 (Discussion)** to better contextualize our empirical results. While absolute improvements over CLIP-LoRA can be modest in certain benchmarks, CLIP-SVD consistently matches or exceeds performance, especially in low-data (1 and 2 shots) and biomedical adaptation settings. Importantly, CLIP-SVD’s contribution is not limited to raw accuracy: by freezing the singular vectors and tuning only the singular values, it better preserves the pretrained representation geometry and provides **spectral-level interpretability**. This interpretability is not inherent to standard low-rank adaptation methods such as LoRA, which learn updates along newly introduced directions (Hu et al., 2021; Shuttleworth et al., 2025). We believe these properties, together with the extensive cross-domain evaluation, constitute a meaningful empirical contribution, and we have adjusted the paper’s framing accordingly.

---

### Decision · Action_Editor_hGLN · 2026-04-07

**Recommendation:** Accept with minor revision

**Additional Comments:**

n.a.

**Audience:**

Yes

**Audience Explanation:**

vision-language fine-tuning community.

**Claims And Evidence:**

Yes

**Claims Explanation:**

This submission provides a thorough and well-executed empirical study of singular-value fine-tuning for CLIP-style vision-language models. While the core idea is not methodologically groundbreaking beyond prior SVF work, the paper makes a meaningful contribution by systematically adapting and evaluating this approach in the multimodal Transformer setting, with strong results across natural and biomedical benchmarks, excellent parameter efficiency, and useful interpretability analysis. The authors addressed the key reviewer concerns regarding novelty framing, computational cost, implementation clarity, and fairness of comparisons. Overall, the paper meets the bar for TMLR as a solid, carefully validated empirical contribution.